# VIDEOPHY: EVALUATING PHYSICAL COMMONSENSE FOR VIDEO GENERATION

**Hritik Bansal**[*1]   **Zongyu Lin**[*1]   **Tianyi Xie**[†1]   **Zeshun Zong**[†1]   **Michal Yarom**[‡2]
**Yonatan Bitton**[‡2]   **Chenfanfu Jiang**[1]   **Yizhou Sun**[1]   **Kai-Wei Chang**[1]   **Aditya Grover**[1]

[1]**University of California Los Angeles**   [2]**Google Research**

## ABSTRACT

Recent advances in internet-scale video data pretraining have led to the development of text-to-video generative models that can create high-quality videos across a broad range of visual concepts, synthesize realistic motions and render complex objects. Hence, these generative models have the potential to become general-purpose simulators of the physical world. However, it is unclear how far we are from this goal with the existing text-to-video generative models. To this end, we present VIDEOPHY, a benchmark designed to assess whether the generated videos follow physical commonsense for real-world activities (e.g. marbles will roll down when placed on a slanted surface). Specifically, we curate diverse prompts that involve interactions between various material types in the physical world (e.g., solid-solid, solid-fluid, fluid-fluid). We then generate videos conditioned on these captions from diverse state-of-the-art text-to-video generative models, including open models (e.g., CogVideoX) and closed models (e.g., Lumiere, Dream Machine). Our human evaluation reveals that the existing models severely lack the ability to generate videos adhering to the given text prompts, while also lack physical commonsense. Specifically, the best performing model, CogVideoX-5B, generates videos that adhere to the caption and physical laws for $39.6\%$ of the instances. VIDEOPHY thus highlights that the video generative models are far from accurately simulating the physical world. Finally, we propose an auto-evaluator, VIDEOCON-PHYSICS, to assess the performance reliably for the newly released models. Code: `https://github.com/Hritikbansal/videophy`.

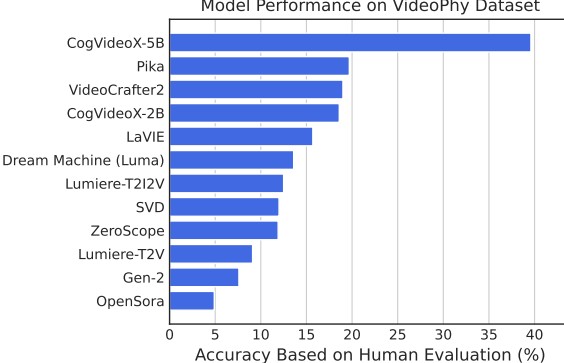

Figure 1: **Model performance on the VIDEOPHY dataset using human evaluation.** We assess the physical commonsense and semantic adherence to the conditioning caption in the generated videos. We find that CogVideoX-5B can generate videos that follow the caption and physics commonsense for $39.6\%$ of the prompts, while the other models are far behind ($< 20\%$). This indicates that the existing models severely lack the ability of being general-purpose physical world simulators.

## 1 INTRODUCTION

Recent advancements in pretraining on internet-scale video data [3, 122, 113, 111, 24] have led to the development of various text-to-video (T2V) generative models such as Sora [68] that can generate photo-realistic videos conditioned on a text prompt [8, 110, 22, 80, 98, 15, 49]. Specifically, these models can generate complex scenes (e.g., 'busy street in Japan') and realistic motions (e.g., 'running', 'pouring'), making them amenable for understanding and simulating the physical world. As humans, we develop an intuitive understanding of the object interactions through our experience

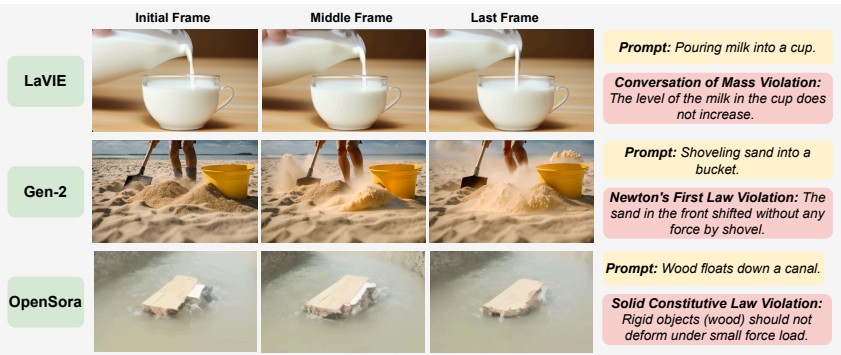

Figure 2: **Illustration of poor physical commonsense by various T2V generative models.** Here, we show that the generated videos can violate a diverse range of laws pf physics such as conversation of mass, Newton's first law, and solid constitutive laws. In VIDEOPHY, we curate a wide range of prompts that would be used to assess the physical commonsense of the T2V models.

with the real-world, without any formal education in physics (also termed as intuitive physics) [27]. For instance, we can predict the trajectories of the billiard balls after an application of force. Recent efforts [26, 19, 1] have further utilized text-guided video generation to train agents that can act, plan, and solve goals in the real world. In spite of the strong physical motivations of these works, it remains unclear *how well the generated videos from T2V models adhere to the laws of physics*.

One might be tempted to assess the physical commonsense of generated videos by comparing them with physical simulations as a ground truth. However, this is non-trivial, and no similar approaches have been proposed yet. The main challenges include the lack of mature methods to accurately generate 3D geometries from single-view images or video, which is essential for physical simulations. Further, physical simulations usually require precise tuning of material parameters based on the expertise of graphics researchers to match real-world dynamics. Recently, some efforts have been made to tune simulation parameters from generated videos (e.g., [40, 64, 130, 75]). Nevertheless, they depend on the physical plausibility of the generated videos themselves, which is again the open question we want to address. Finally, the accurate lighting and rendering are also necessary to convert physically simulated results into images and videos, yet these parameters are also unknown. Most importantly, it should be noted that physical simulations are not equivalent to ground truth. They are merely numerical solvers of differential equations that attempt to approximate and describe real-world dynamics based on models proposed by researchers. Prior work such as VBench [42, 67] introduced a comprehensive benchmark to evaluate various qualities of generated videos (e.g., motion smoothness, background consistency) using existing models, but it does not specifically address the generated videos' adherence to physical laws. Therefore, existing benchmarks and metrics are either unreliable or lack coverage for holistic evaluation of the physical commonsense capabilities.

To this end, we propose VIDEOPHY, a dataset designed to evaluate the adherence of generated videos to physical commonsense in real-world scenarios. Specifically, we focus on the intuitive understanding of the behavior and dynamics of various states of matter (solids, fluids) in the physical world [84, 126, 11]. For instance, 'water pouring into a glass' will intuitively result in the water level in the glass rising over time. As a result, we rely on human perception and experience in the physical world to assess the adherence of the generated videos to physical laws instead of precise dynamical equations, which are harder to assess. In Figure 2, we provide qualitative examples to illustrate physical commonsense violations in the videos. Our dataset is constructed through a three-stage pipeline that involves (a) prompting a large language model [78] to generate candidate captions that depict interactions between diverse states of matter (e.g., solid-solid, solid-fluid, fluid-fluid), (b) human verification of the generated captions, and (c) annotating the complexity in rendering objects or synthesizing motions described in the captions based on physics simulation.

In total, VIDEOPHY comprises 688 high-quality, human-verified captions that will be used to generate videos from T2V models. In addition, the dataset consists human-labeled annotations for physical commonsense of the generated videos. Specifically, we acquire generated videos from **twelve** diverse T2V models including open models (e.g., OpenSora [80], SVD [13], VideoCrafter2 [22],

CogVideoX [123]) and closed models (e.g., Pika [83], Lumiere [8], Gen-2 [29], Dream Machine [2]). Subsequently, we perform human evaluation on the generated videos for semantic adherence to the conditioning text (e.g., do the videos follow the caption?) and physical commonsense (e.g., do the videos follow physical laws intuitively?). Interestingly, we find that the existing T2V generative models severely lack the capability to follow caption accurately and generate videos with physical commonsense. Specifically, the best performing model, CogVideoX-5B, follows the text and generates physically accurate videos for 39.6% of the instances (§5). Our fine-grained analysis reveals that current T2V models are particularly poor at generating physically plausible videos for prompts that require solid-solid interaction (e.g., ball bouncing on the floor, hammer hits a nail). In Figure 1, we compare the performance (i.e., accurate semantic adherence and physical commonsense) of various T2V generative models on the VIDEOPHY dataset. In addition, we perform a detailed qualitative analysis to study the modes of the failures for different models in detail (§5.2). In particular, we observe that the models often struggle to accurately identify individual objects and comprehend their material properties, which is essential for generating physically plausible dynamics. For instance, an object recognized as a rigid body in the physical world should not deform over time.

Although human evaluation of semantic adherence and physical commonsense is reliable, it is both expensive and difficult to scale. To address this challenge, we introduce VIDEOCON-PHYSICS, an open video-language model designed to assess the semantic adherence and physical commonsense of generated videos using user queries grounded in text (§6). Specifically, we finetune VIDEOCON [5], a robust semantic adherence evaluator for real videos, on generated videos and human annotations collected as a part of our dataset. Our results demonstrate that VIDEOCON-PHYSICS outperforms Gemini-Pro-Vision-1.5 [90], showing a 9 points improvement in semantic adherence and a 15 points improvement in physical commonsense on unseen prompts. Further, we show that VIDEOCON-PHYSICS generalizes to unseen generative models, which established its reliability for evaluating future generative models. Overall, the VIDEOPHY dataset aims to bridge the gap in understanding physical commonsense in generated videos and enables scalable testing for upcoming T2V models.

## 2 VIDEOPHY DATASET

Our dataset, VIDEOPHY, aims to offer a robust evaluation benchmark for physical commonsense in video generative models. Specifically, the dataset is curated with guidelines to cover (a) a wide range of daily activities and objects in the physical world (e.g., rolling objects, pouring liquid into a glass), (b) physical interactions between various material types (e.g., solid-solid or solid-fluid interactions), and (c) the perceived complexity of rendering objects and motions under graphic simulation. For instance, *ketchup*, which follows non-newtonian fluid dynamics [114], is harder to model and simulate than *water*, which follows newtonian fluid dynamics, using traditional fluid simulators [16]. Under the collection guidelines, we curate a list of text prompts that will be used for conditioning the text-to-video generative models. Specifically, we follow the 3-stage pipeline to create the dataset.

| Category | Difficulty | Example Captions |
|---|---|---|
| Solid-Solid | Easy | Bottle topples off the table. ( rigid bodies ) |
| Solid-Solid | Hard | Scrubber scrubs a dirty dish. ( complex contacts ) |
| Solid-Fluid | Easy | Water flows down a circular drain. ( contacts with rigid bodies ) |
| Solid-Fluid | Hard | A swimmer splashing in the sea water. ( contacts with high-speed ) |
| Fluid-Fluid | Easy | Rain splashing on a pond. ( mixing of same fluids ) |
| Fluid-Fluid | Hard | Ink spreading in still water. ( mixing of different fluids ) |

Table 1: **Example captions in the VIDEOPHY dataset.** Specifically, we design them to depict the interactions between two states of matter (solid-solid, solid-fluid, fluid-fluid). We further classify the captions as easy or hard based on the modeling and simulation complexity in the computer graphics. We highlight the reasoning behind the easy and hard annotations by our expert annotators in the () .

**LLM-Generated Captions (Stage 1).** Here, we query a large language model, in our case GPT-4 [78], to generate a list of 1000 *candidate* captions depicting real-world dynamics. As the majority of real-world dynamics involve solids or fluids, we broadly classify those dynamics into three categories:

Table 2: Statistics of the VIDEOPHY dataset.

| Statistic | Number |
|---|---|
| Total captions | 688 |
| Unique actions | 138 |
| Total T2V models | 12 |
| Total generated videos | 11330 |
| Human annotations | 36500 |
| Category (Interacting materials) | 3 |
| Solid-Solid | 289 |
| Solid-Fluid | 291 |
| Fluid-Fluid | 108 |
| Category (Interaction complexity) | 2 |
| Easy | 366 |
| Hard | 322 |

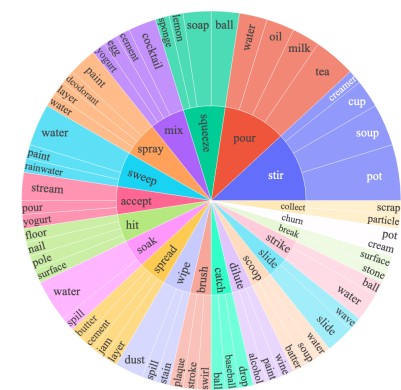

Figure 3: Top-20 most frequently occurring verbs (inner) and their top-4 direct nouns (outer) in captions.

*solid-solid* interactions, *solid-fluid* interactions, and *fluid-fluid* interactions. Specifically, we consider fluid dynamics involving in-viscid and viscous flows—representative examples being water and honey, respectively. On the other hand, we find that solids exhibit more diverse constitutive models, including but not limited to rigid bodies, elastic materials, sands, metals, and snow. In total, we prompt GPT-4 to generate 500 candidate captions for solid-solid and solid-fluid interactions, and 200 candidate captions for fluid-fluid interactions. We present the GPT-4 prompts in Appendix G.

**Human Verification (Stage 2).** Since LLM-generated captions may not adhere to our input query, we perform a human verification step to filter bad generations. Specifically, the authors perform human verification to ensure the quality and relevance of the captions, adhering to these criteria: (1) the caption must be clear and understandable; (2) the caption should avoid excessive complexity, such as overly varied objects or too intricate dynamics; and (3) the captions must accurately reflect the intended interaction categories (e.g., that fluids are mentioned in solid-fluid or fluid-fluid dynamics). Finally, we have 688 captions where 289 captions for solid-solid interactions, 291 for solid-fluid interactions, and 108 for fluid-fluid interactions, respectively. We highlight that our prompts include a wide range of material types and physical interactions that are common in both real life and the graphics community. Material types include simple rigid bodies [60], deformable bodies [41], think shells [23], metal [59], fracture [115], cream [129], sand [47] and so on. The contact handling is also diverse as it is based on the interactions of all aforementioned materials [35, 56, 36, 133]. We provide more discussion on categorization in Appendix B. We highlight that the data quality is paramount for evaluating foundation models. For instance, Winoground (400 examples) [105], Visit-Bench (500 examples) [12], LLaVA-Bench (90 examples) [65], and Vibe-Eval (269 examples) [81] are commonly employed to assess vision-language models due to their high-quality despite their limited size. Given that human verification demands significant expert hours and is not scalable within our budget, we prioritize data quality for evaluating T2V models.

**Difficulty Annotation (Stage 3).** To acquire fine-grained insights into the quality of the video generation, we further annotate our each instance in the dataset with perceived *difficulty*. Specifically, we ask two experienced graphics researchers (senior Ph.D. students in physics-based simulation) to independently classify each caption as easy (0) or hard (1) based on their perception of the complexity in simulating the objects and motions in the captions using state-of-the-art physics engines [56, 25, 118, 132, 87, 32]. Subsequently, the disagreements were discussed to reach a unanimous judgment for less than 5% of the instances. The difficulty of a simulation is primarily influenced by the complexity of the model, which varies depending on the type of material. For example, deformable bodies pose a greater modeling challenge than rigid bodies because they change shape under external forces, leading to more complex partial differential equations (PDEs). In contrast, rigid bodies maintain their shape, resulting in simpler models. Another key factor is the numerical difficulty in solving these equations, which increases with the material's velocity, especially when high-order terms are involved in the PDEs. As a result, slower-moving materials are generally easier to simulate than faster-moving ones. We note that the level of difficulty is evaluated within

each category (e.g., solid-solid, solid-fluid, fluid-fluid), and cannot be compared across different categories. We present the examples for generated captions in Table 1.

**Data Analysis.** A fine-grained metadata facilitates a comprehensive understanding of the benchmark. Specifically, we present the main statistics of the VIDEOPHY dataset in Table 2. Notably, we generate 11330 videos for the prompts in the dataset using a diverse range of generative models. In addition, the average caption length is 8.5 words, indicating that most captions are straightforward and do not complicate our analysis with complex phrasing that could be excessively challenging the generative models. [1] The dataset includes 138 unique actions grounded in our captions. Figure 3 visualizes the root verbs and direct nouns used in the VIDEOPHY captions, highlighting the diversity of actions and entities. Hence, our dataset encompasses a wide range of visual concepts and actions. We perform fine-grained diversity analysis in Appendix J.

## 3 EVALUATION

### 3.1 METRICS

The ability to assess the quality of the generated videos is a challenging task. While humans can evaluate videos across various visual dimensions [42, 20], we focus primarily on the models' adherence to the provided text and the incorporation of physical commonsense. These are key objectives that conditional generative models must maximize. We note that several video characteristics such as object motion, video quality, text adherence, physical commonsense, temporal consistency of subject and object etc. are usually intertwined with each other. It is non-trivial to disentangle their effect when humans make decisions. However, focusing on each aspect at a time provides a comprehensive picture of the model capabilities along a specific dimension. In this work, we focus on physical commonsense and semantic adherence. Further, there are diverse ways to acquire human judgments such as dense and sparse feedback. While a dense feedback provides detailed information about the model mistakes, it is hard to acquire and miscalibrated [69, 55]. Due to the simplicity of binary judgment and its widespread use in text-to-image generative models [58, 54], we employ binary feedback (0/1) to evaluate the generated videos in this work (more discussion in Appendix C). Further, our experiments will demonstrate that binary feedback effectively highlights differences in the model's quality across various object interactions and levels of task complexity.

**Semantic Adherence (SA).** This metric assesses whether the text caption is semantically grounded in the frames of the generated videos, measuring video-text alignment. Specifically, it assesses if the actions, events, entities, and their relationships are perceived to be correctly depicted in the video frames (e.g., water is flowing into the glass in the generated video for the caption 'water pouring into the glass'). In this work, we annotate the generated videos for semantic adherence, denoted as $SA = \{0, 1\}$. Here, $SA = 1$ indicates that the caption is grounded in the generated video.

**Physical Commonsense (PC).** This metric evaluates whether the depicted actions, and object's state follow the physics laws in the real-world. For instance, the level of water should increase in the glass as water flows into it, following conversation of mass. In this work, we annotate the physical commonsense of the generated videos, denoted as $PC = \{0, 1\}$. Here, $PC = 1$ indicates that the generated movements and interactions align with intuitive physics that humans acquire with their experience in the real-world. As physical commonsense is entirely grounded in the video, it is independent of the semantic adherence capability of the generated video. In this work, we compute the fraction of the videos for which semantic adherence is high ($SA = 1$), physical commonsense is high ($PC = 1$), and joint performance of these metrics is high ($SA = 1, PC = 1$).

### 3.2 HUMAN EVALUATION

We conducted a human evaluation to assess the performance of the generated videos in terms of semantic adherence and physical commonsense using our dataset. The annotations were obtained from a group of qualified Amazon Mechanical Turk (AMT) workers who were provided with the detailed task description (and clarifications) on a shared slack channel. Subsequently, 14 workers who have

---

[1]We use GPT-4 to enhance and generate longer versions of the original captions. However, we found that most of the T2V models are poor at following long/enhanced captions most of the time.

studied high-school physics were chosen to perform the annotations after passing a qualification test. In this task, annotators were presented with a caption and the corresponding generated video without any information about the generative model. They were asked to provide a semantic adherence score (0 or 1) and a physical commonsense score (0 or 1) for each instance. Annotators were instructed to treat semantic adherence and physical commonsense as independent metrics and were shown several solved examples by the authors before starting the main annotation task. In some cases, we find that generative models create static scenes instead of video frames with high motion. Here, we ask annotators to judge the physical plausibility of the static scene in the real world (e.g., a static scene of a folded brick does not follow physical commonsense). If the static scenes are noisy (e.g., unwanted grainy or speckled patterns), we instruct them to consider it as poor physical commonsense. [2]

The human annotators were not asked to list the violation of the physics laws since it would make the annotations more time-consuming and expensive. Additionally, the current annotations can be performed by annotators experience in the physical world (e.g., workers know that water flows *down* from a tap, shape of a wood log *will not change* while floating on water) instead of advanced education in physics. A screenshot of the human annotation interface is presented in Appendix H.

### 3.3 AUTOMATIC EVALUATION

While the human evaluation is more accurate for benchmarking, it is time-consuming and expensive to acquire at scale. In addition, we want the model developers with limited resources for human evaluation to use our benchmark. To this end, we design **VIDEOCON-PHYSICS**, a reliable auto-rater for our evaluation dataset. Specifically, we use VIDEOCON, an open video-text language model with 7B parameters, that is trained on real videos for robust semantic adherence evaluation [5]. Specifically, we prompt VIDEOCON to generate a text response (*Yes/No*) conditioned on the multimodal template. We provide details about the templates and score computation using VIDEOCON in Appendix I.

Since VIDEOCON is not trained on the generated video distribution or equipped to judge physical commonsense, it is not expected to perform well in our setup in a zero-shot manner. Prior work [74] has shown that data-driven approaches can outperform rule-based physics simulators for more complicated systems like weather and climate. Hence, we take a data-driven approach in this work. To this end, we propose VIDEOCON-PHYSICS, an open-source generative video-text model, that can assess the semantic adherence and physical commonsense of the generated videos. Specifically, we finetune VIDEOCON by combining the human annotations acquired for the semantic adherence and physical commonsense tasks over the generated videos. We present the GPT-4V [79] and Gemini-1.5-Pro-Vision [90] baselines in Appendix M.[3] We assess auto-rater effectiveness by computing the ROC-AUC between humans and its judgments for videos generated with testing prompts.

### 4 SETUP

**Video Generative Models.** We evaluate a diverse range of **twelve** closed and open text-to-video generative models on VIDEOPHY dataset. The list of the models includes *ZeroScope* [21], *LaVIE* [112], *VideoCrafter2* [22], *OpenSora* [80], CogVideoX-2B and 5B [123], *StableVideoDiffusion (SVD)-T2I2V* [13], *Gen-2 (Runway)* [29], *Lumiere-T2V*, *Lumiere-T2I2V* (Google) [8], Dream Machine (Luma AI) [2], and *Pika* [83]. We provide more model and inference details in Appendix F and N. [4]

**Dataset setup.** As described earlier, we train VIDEOCON-PHYSICS to enable cheaper and scalable testing of the generated videos on our dataset (§ 3.3). To facilitate this, we split the prompts in the VIDEOPHY dataset equally into *train* and *test* sets. Specifically, we utilize the human annotations on the generated videos for the 344 prompts in the *test* set for benchmarking, while the human annotations on the generated videos for the 344 prompts in the *train* set are used for training the automatic evaluation model. We ensure that the distribution of the state of matter (solid-solid, solid-fluid, fluid-fluid) and complexity (easy, hard) is similar in the training and testing.

---

[2] The workers were compensated at a rate of $18 per hour.

[3] We note that finetuning separate classifier for semantic adherence and physical commonsense did not provide any additional benefits over a single classifier (VIDEOCON-PHYSICS) trained in a multi-task manner.

[4] While there are various closed models such as Sora [68], Kling AI [48], and Genmo [34], we could not get access through their videos due to the lack of API support.

Table 3: **Human evaluation results on the VIDEOPHY dataset.** We report the percentage of testing prompts for which the T2V models generate videos that adhere to the conditioning caption and exhibit physical commonsense. We abbreviate semantic adherence as SA, and physical commonsense as PC. **SA, PC** indicates the percentage of the instances for which SA=1 and PC=1. Ideally, we want the generative models to maximize the performance on this metric. In the first column, we highlight the overall performance, and the later columns are dedicated to fine-grained performance for the interaction between different states of matter in the prompts.

| Model | Overall (%) | | | Solid-Solid (%) | | | Solid-Fluid (%) | | | Fluid-Fluid (%) | | |
|---|---|---|---|---|---|---|---|---|---|---|---|---|
| | SA, PC | SA | PC | SA, PC | SA | PC | SA, PC | SA | PC | SA, PC | SA | PC |
| *Open Models* | | | | | | | | | | | | |
| CogVideoX-5B [123] | 39.6 | 63.3 | 53 | 24.4 | 50.3 | 43.3 | 53.1 | 76.5 | 59.3 | 43.6 | 61.8 | 61.8 |
| VideoCrafter2 [22] | 19.0 | 48.5 | 34.6 | 4.9 | 31.5 | 23.8 | 27.4 | 57.5 | 41.8 | 32.7 | 69.1 | 43.6 |
| CogVideoX-2B [123] | 18.6 | 47.2 | 34.1 | 12.7 | 42.9 | 28.1 | 21.9 | 56.1 | 34.9 | 25.4 | 34.5 | 47.2 |
| LaVIE [112] | 15.7 | 48.7 | 28.0 | 8.5 | 37.3 | 19.0 | 15.8 | 52.1 | 30.8 | 34.5 | 69.1 | 43.6 |
| SVD-T2I2V [14] | 11.9 | 42.4 | 30.8 | 4.2 | 25.9 | 27.3 | 17.1 | 52.7 | 32.9 | 18.2 | 58.2 | 34.5 |
| ZeroScope [21] | 11.9 | 30.2 | 32.6 | 6.3 | 17.5 | 22.4 | 14.4 | 40.4 | 37.0 | 20.0 | 36.4 | 47.3 |
| OpenSora [80] | 4.9 | 18.0 | 23.5 | 1.4 | 7.7 | 23.8 | 7.5 | 30.1 | 21.9 | 7.3 | 12.7 | 27.3 |
| *Closed Models* | | | | | | | | | | | | |
| Pika [83] | 19.7 | 41.1 | 36.5 | 13.6 | 24.8 | 36.8 | 16.3 | 46.5 | 27.9 | 44.0 | 68.0 | 58.0 |
| Dream Machine [2] | 13.6 | 61.9 | 21.8 | 12.1 | 50.0 | 24.3 | 16.6 | 68.1 | 23.6 | 9.0 | 76.3 | 11.0 |
| Lumiere-T2I2V [8] | 12.5 | 48.5 | 25.0 | 8.4 | 37.1 | 25.2 | 17.1 | 59.6 | 26.0 | 10.9 | 49.1 | 21.8 |
| Lumiere-T2V [8] | 9.0 | 38.4 | 27.9 | 8.4 | 26.6 | 27.3 | 9.6 | 47.3 | 26.0 | 9.1 | 45.5 | 34.5 |
| Gen-2 [29] | 7.6 | 26.6 | 27.2 | 4.0 | 8.9 | 37.1 | 8.1 | 38.5 | 18.5 | 15.1 | 37.7 | 26.4 |

**Benchmarking.** Here, we generate one video per test prompt for each T2V generative model in our testbed. Subsequently, we ask three human annotators to judge the semantic adherence and physical commonsense of the generated videos. In our experiments, we report the majority-voted scores from the human annotators. We find that the inter-annotator agreement for semantic adherence and physical commonsense judgment is $75\%$ and $70\%$, respectively. This indicates that the human annotators find the task of judging physical commonsense more subjective than semantic adherence. [5] In total, we collect 24500 human annotations across the testing prompts and T2V models.

**Training set for VIDEOCON-PHYSICS.** Here, we sample two videos per training prompt for nine T2V models.[6] We choose two videos to obtain more data instances for training the automatic evaluation model. Subsequently, we ask one human annotator to judge the semantic adherence and physical commonsense of the generated videos. In total, we collect 12000 human annotations, half of them for semantic adherence and the other half for physical commonsense. Specifically, we finetune VIDEOCON to maximize the log likelihood of *Yes/No* conditioned on the multimodal template for semantic adherence and physical commonsense tasks (Appendix I). We do not collect three annotations per video as it is financially expensive. In total, we spent $3500 on collecting human annotations for benchmarking and training.

## 5 RESULTS

### 5.1 PERFORMANCE ON VIDEOPHY DATASET

We compare the performance of the T2V generative models on the VIDEOPHY dataset using human evaluation in Table 3. We find that CogVideoX-5B generates videos that adhere to the caption and follow physics laws (SA = 1, PC = 1) in $39.6\%$ of the cases. The success of CogVideoX can be attributed to its high-quality data curation including inclusion of detailed captions, and filtering videos with less motion and poor quality. In addition, we find that the rest of the video models achieve a score below $20\%$. This highlights that the existing video models severely lack the capability to generate videos which follow intuitive physics, and establishes VIDEOPHY as a challenging dataset.[7]

More specifically, CogVideoX-5B stands out as the best model for generating videos that demonstrate physical commonsense, achieving a performance of $53\%$, while CogVideoX-2B is the second best

---

[5] Variations in annotations arise from differing tolerance for commonsense violations in imperfect videos. As generative models improve, human annotations will align more closely.

[6] Since the CogVideoX and Dream Machine models were released very recently, they could not be included in the training set of the automatic evaluator.

[7] We compared pairwise model predictions using the paired t-test at a $95\%$ confidence interval. We find that the difference between CogVideoX-5B and other video models is statistically significant (p<0.0001).

open model at $34.1\%$. Further, this highlights that scaling the network capacity improves its ability to capture the underlying physical constraints of the internet-scale video data. In addition, we find that OpenSora performs the worst on the VIDEOPHY dataset, indicating significant potential for the community to improve open-source implementations of Sora. Amongst the closed models, Pika achieves generates videos that achieve positive judgement for semantic adherence and physical commonsense for $19.7\%$ of the cases. Interestingly, we observe that Dream Machine achieves a high semantic adherence score ($61.9\%$) but a poor physical commonsense score ($21.8\%$) which highlights that a optimizing for semantic adherence does not necessarily lead to good physical commonsense.

**Variation with the states of matter.** We study the variation in the performance of T2V models with the interaction between the diverse states of matter grounded in the captions (e.g., solid-solid) in Table 5.1. Interestingly, we find that all the existing T2V models perform the worst on the captions that depict interactions between solid materials (e.g., *bottle* topples off the *table*), with the best performing model, CogVideoX-5B, achieving $24.4\%$ on accurate semantic adherence and physical commonsense. Furthermore, we observe that Pika achieves the highest performance in the captions that depict interaction between fluid and fluid material types (e.g., rain splashing on a pond). This indicates that the T2V model performance is greatly influenced by the states of matter involved in a scene, and highlights that model developers can focus on enhancing semantic adherence and physical commonsense for solid-solid interactions.

**Variation with the complexity.** We analyze the variation in the video model performance with the complexity in rendering objects or synthesizing interactions grounded in the caption under physical simulation in Appendix Table 6. We find that the semantic adherence and physical commonsense performance of all the video models decreases as the complexity of the captions increases. This indicates that the captions that are harder to simulate physically are also harder to control via conditioning for the video generative models. Our analysis thus highlights that the future T2V model development should focus on reducing the gap between the easy and the hard captions from our VIDEOPHY dataset. We provide qualitative generated examples from captions of varying complexity and material states in Appendix V. Further, we present results for additional metrics in Appendix L.

**Correlation analysis.** To understand the connection between various performance metrics, we examine the correlation between semantic adherence (SA) and physical commonsense (PC) with video quality and motion (Appendix §S). Our empirical results show a positive correlation between video quality and both PC and SA, while motion exhibits a negative correlation with PC and SA. This indicates that the video models tend to make more mistakes in the SA and PC when more motions are depicted in them. The closed models (Dream Machine/Pika) contribute to the higher end of the video quality while open models (Zeroscope/OpenSora) contribute to the lower end of video quality. While the high quality is 'correlated' with the better PC, we note that the absolute performance of the models is quite poor on our benchmark.

## 5.2 QUALITATIVE ANALYSIS

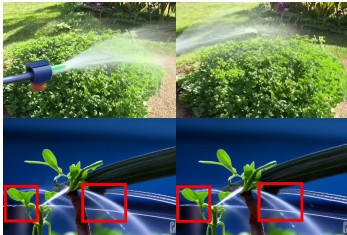 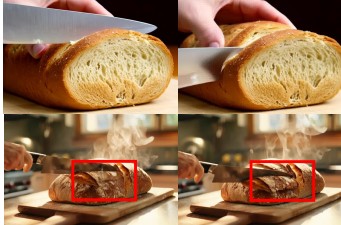 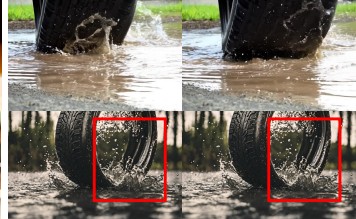

(a) v. Pika. *Water spraying from a garden hose onto plants.*

(b) v. Dream Machine. *The sharp knife severs the fresh loaf of bread.*

(c) v. Gen-2. *A tire rolls through a deep puddle, splashing water.*

Figure 4: **Comparison of CogVideoX-5B with other models.** The top row shows the videos generated by CogVideoX-5B. (a) For Pika, the water streams on the left and right have drastically different speed. (b) For DM, a part of the bread suddenly changes its shape. (c) For Gen-2, the water droplets remain still in the air.

Here, we provide a qualitative analysis of the generated videos to assess the common failure modes.

(a) *Dominoes toppling one after another on the table*   (b) *Leather glove catching a hard baseball*

Figure 5: **Illustration of CogVideoX-5B's limitations in understanding material properties.** Even the best-performing model, CogVideoX-5B, may struggle to correctly capture the material properties, leading to unnatural dynamics that do not align with the object characteristics. Artifacts in the examples: (a) the dominoes, which should behave as rigid bodies, show inconsistent changes in geometry and texture over time, (b) the leather glove exhibits unnatural deformations.

**Comparison between CogVideoX-5B with other models.** We analyze some qualitative examples to understand the gap between the best-performing model (CogVideoX-5B) and the other models in our testbed. We present some examples in Figure 4. Specifically, we find that SVD-T2I2V is likely to underperform in scenes involving vibrant fluid dynamics. Lumiere-T2I2V and Dream Machine (Luma) perform better than Lumiere-T2V in terms of visual quality, but they lack profound understanding of rigid geometries (e.g. in Figure 4(b)). Further, we notice that Gen-2 sometimes generates static objects in the air with slow camera motion, instead of meaningful physical dynamics (e.g. in Figure 4(c)). In contrast, CogVideoX-5B shows decent capability of identifying distinct objects, as deformation from its results seldom mingles multiple objects. Further, it tends to use simpler backgrounds, avoiding complex patterns where flaws are easier to be spotted. Nevertheless, even the best-performing model, CogVideoX-5B, may struggle to understand the material properties of the underlying objects, resulting in unnatural or inconsistent deformations, as shown in Figure 5. This phenomenon is also observed in results from other video generative models. Our analysis highlights the lack of fine-grained physical commonsense that future research should aim to address.

**Failure mode analysis.** We present some qualitative examples to understand the common failure modes in the generated video regarding poor physical commonsense. Qualitative examples from various T2V generative models are provided in Figure 15 - 26 in Appendix U. The common failure modes include – (a) *Conservation of mass violation*: the volume or texture of an object is not consistent over time, (b) *Newton's First Law violation*: an object changes its velocity in a balanced state without any external force, (c) *Newton's Second Law violation*: an object violates the conversation of momentum, (d) *Solid Constitutive Law violation*: solids deform in ways that contradict their material properties, e.g., a rigid object deforming over time, (e) *Fluid Constitutive Law violation*: fluids exhibit unnatural flow motions, and (f) *Non-physical penetration*: objects unnaturally penetrate each other.

Table 4: **Comparison of ROC-AUC for automatic evaluation methods.** We find that VIDEOCON-PHYSICS outperforms diverse baselines, including GPT-4Vision and Gemini-1.5-Pro, for semantic adherence (SA) and physical commonsense (PC) judgments on the testing prompts.

| Method(↓)/RUC-AOC(→) | SA | PC |
|---|---|---|
| Random | 50 | 50 |
| GPT-4-Vision [79] | 53 | 53 |
| Gemini-1.5-Pro-Vision [90] | 73 | 58 |
| VIDEOCON [5] | 65 | 54 |
| VIDEOCON-PHYSICS (Ours) | 82 | 73 |

## 6 VIDEOCON-PHYSICS: AUTOMATIC EVALUATOR FOR VIDEOPHY DATASET

We supplement our dataset with VIDEOCON-PHYSICS, an automatic rater for scalable and reliable evaluation of semantic adherence and physical commonsense in the generated videos.

**VIDEOCON-PHYSICS generalizes to unseen prompts.** We compare the ROC-AUC of different automatic evaluators with the human predictions on the testing prompts in Table 4. Here, the videos are generated by the models that are used to train the VIDEOCON-PHYSICS model. We find that the VIDEOCON-PHYSICS outperforms the zero-shot VIDEOCON by 17 points and 19 points on the semantic adherence and physical commonsense judgment, respectively. This highlights that finetuning with the generated video distribution and human annotations aids in improving the model judgment on

the unseen prompts. Further, we notice that the model's agreement are higher for semantic adherence as compared to the physical commonsense. This indicates that judging physical commonsense is a harder task than judging semantic adherence for VIDEOCON-PHYSICS. Interestingly, we observe that the GPT-4-Vision's judgments are close to random for semantic adherence and physical commonsense on our dataset. This implies that faithful evaluations are hard to obtain from the multi-image reasoning capabilities of the GPT-4-Vision in a zero-shot manner. To address this, we test Gemini-Pro-Vision-1.5 and find that it achieves a good semantic adherence score (73 points), however, it is close to random in physical commonsense evaluation (54 points). This highlights that the existing multimodal foundation models lack the capability to judge physical commonsense.

**VIDEOCON-PHYSICS generalizes to unseen generative models.** To assess performance on an unseen video distribution, we train an ablated version of VIDEOCON-PHYSICS on a restricted set of video data. Specifically, we train VIDEOCON-PHYSICS on human annotations acquired from VideoCrafter2, ZeroScope, LaVIE, OpenSora, SVD-T2I2V and Gen-2, and evaluate it on unseen videos from the remaining T2V models in our testbed generated for the testing captions. We present the results in Table 5. We find that VIDEOCON-PHYSICS outperforms

Table 5: **Performance of VIDEOCON-PHYSICS on unseen generative model.** We train an ablated version of VIDEOCON-PHYSICS and find that it outperforms the baseline in the semantic adherence (SA) and physical commonsense (PC) judgment averaged over three unseen video models on the testing prompts.

| Method | SA | PC |
|---|---|---|
| VIDEOCON [5] | 64 | 57 |
| VIDEOCON-PHYSICS (Ours) | 79 | 72 |

VIDEOCON by 15 points and 15 points on semantic adherence and physical commonsense judgement, respectively. This highlights that VIDEOCON-PHYSICS can judge semantic adherence and physical commonsense as new T2V generative models are released.

**Automatic leaderboard reliably tracks human leaderboard.** We create an automatic leaderboard by averaging the semantic adherence and physical commonsense scores of the open and closed video models on the test set. Subsequently, we align these rankings with the human leaderboard based on the joint performance metrics ($SA = 1, PC = 1$). We present the human and automatic leaderboard for the open and closed model in Appendix P. We observe that the relative rankings of the models in the automatic leaderboard (CogVideoX-5B>VideoCrafter2>LaVIE>CogVideoX-2B>SVD-T2I2V>ZeroScope>OpenSora) strongly matches with the relative rankings of the models in the human leaderboard (CogVideoX-5B>VideoCrafter2>CogVideoX-2B>LaVIE>SVD-T2I2V>ZeroScope>OpenSora). We observe similar trends for the closed models. But, we find that Pika achieves a relatively low score on the automatic leaderboard, a limitation that can be improved by acquiring more data for VIDEOCON-PHYSICS. Overall, we find that the rankings of most of the models are similar under both the leaderboards, establishing its reliability for future model development. Further discussion on the usefulness of VIDEOCON-PHYSICS in Appendix §R.

**Finetuning video models.** While VIDEOPHY data is used for model evaluation and building automatic evaluator, we assess whether this dataset can be used to finetune video models in Appendix T. Post-finetuning, we observe a significant decrease in semantic adherence, while physical commonsense remains unchanged. This is likely due to limited training samples, optimization challenges, and the nascency of the video finetuning field. Future work will focus on enhancing physical commonsense in generative models based on these findings.

## 7 CONCLUSION

In this work, we introduce VIDEOPHY, a first of its kind dataset to assess the physical commonsense in the generated videos. Further, we evaluate a diverse set of video models (open and closed models) and found that they significantly lack in the physical commonsense and semantic adherence capabilities. Our dataset unveils that the existing methods are far being general-purpose world simulators. Further, we introduce VIDEOCON-PHYSICS, an auto-evaluation model that enables cheap and scalable evaluation on our dataset. We believe that our work will serve as the cornerstone in studying physical commonsense for video generative modeling.

## REPRODUCIBILITY STATEMENT

In this work, we provide a detailed description about the dataset construction in §4. Specifically, we mention the prompts used for initial caption generation in Appendix G. Further, we provide the details about all the video generative models in §4, along with the inference details in Appendix N. In addition, we provide the details about finetuning VIDEOCON-PHYSICS in Appendix O. Finally, we commit to releasing the data, generated videos, and trained VIDEOCON-PHYSICS in the camera-ready version.

## ACKNOWLEDGEMENT

HB is supported in part by AFOSR MURI grant FA9550-22-1-0380. YS was partially supported by NSF 2211557, NSF 2119643, NSF 2303037, NSF 2312501, SRC JUMP 2.0 Center, Amazon Research Awards, and Snapchat Gifts. KC was partially supported by ONR grant N00014-23-1-2780, and U.S. DARPA ECOLE Program No. #HR00112390060.

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

# A    RELATED WORK

**Video Generation Models.**    Recent advancements in video generation models have emerged from two primary architectures: diffusion-based models [29, 14, 68, 15, 112, 110, 22, 45, 62] and autoregressive modeling-based approaches [127, 49, 38, 107]. Among these, diffusion models have garnered significant attention. The model known as SVD [14], built on a Latent Diffusion Model (LDM) [91], proposes a three-stage training process for video LDMs: text-to-image pretraining, video pretraining, and video finetuning. Sora [68] represents a state-of-the-art in video generation, utilizing a diffusion-transformer architecture with unified training recipes and enhancements in language description processing for video generation. ModelScope [110] is also a diffusion-based text-to-video model which combines a VQGAN [28], a text-encoder, and a denoising UNet. Another diffusion model, VideoCrafter2 [22], leverages low-quality videos and high-quality videos to generate high-quality videos. LaVIE [112] is composed of a base text-to-video model, a temporal interpolation model, and a video super-resolution model, indicating that joint image and video training and temporal self-attention with rotary positional embeddings are key components to boost performance. Given the rapid development of video generation technology, an effective evaluation method for the generated videos becomes crucial. Our paper focuses on evaluating text-to-video generation models for their physical commonsense capabilities.

**Evaluating Video Generation Models.**    To evaluate the quality of a T2V generative model, Fréchet video distance (FVD) is traditionally used to measure the similarity between real and generated video distributions [106, 20]. However, FVD has several limitations for assessing physical commonsense including the requirement for a reference video that is difficult to obtain for novel scenes, bias towards video quality, and failure to detect unrealistic motions [17, 99]. Similarly, CLIPScore [88] measures *semantic* similarity between generated video frames and the conditioning text in a shared representation space, making it unsuitable for evaluating physical commonsense in generated videos.

However, there is a growing consensus on the need for more comprehensive metrics to assess the performance of video generation models [42, 67, 51, 61]. V-Bench [42] offers a detailed benchmark suite that introduces a hierarchical evaluation protocol, breaking down 'video generation quality' into various granular perspectives. Another framework, EvalCrafter [67], proposes 17 objective metrics. Despite these advancements, existing methods largely overlook the fundamental aspect of physical commonsense. Unlike static images, videos incorporate a temporal dimension, embedding physical commonsense information across frames. Our research dives into the measurement of physical commonsense [11] in videos. Additionally, we introduce a VIDEOCON-PHYSICS auto-evaluator and analyze specific physical laws that are violated in the generated videos through qualitative analysis.

**Physics Modeling.**    Simulating physical behaviors of solids and fluids has always been an important and popular topic in computer graphics. For solid materials, the simplest physical model is the long-established rigid body simulation [9], where solids are assumed not to deform. Simulation of deformable solids [97], on the other hand, takes into account the strain and stress during deformation. To capture more complicated materials, researchers have been proposing increasingly intricate models for different materials, such as metal [77], sand [47], and snow [101]. In contrast, most of the common fluids [16] in daily life can be broadly categorized as inviscid [50], e.g., water and air, and viscous fluids [103, 53], e.g., honey and oil. Additionally, an orthogonal research direction is to accurately, efficiently, and robustly model contact and interaction between different materials. These include solid-solid [56, 57], solid-fluid [10, 118], and fluid-fluid interactions [73]. Further, recent advancements in computer vision have started exploring incorporating physics priors into various 3D-aware generation tasks to enhance physical plausibility, such as human animation [128, 95, 121] and 3D/4D generation [71, 119, 130]. However, these approaches often depend on high-quality 3D reconstructions from multi-view images. Some efforts [66] have also integrated physics-based simulations into video generative models, but the simulations are performed in 2D space due to the lack of 3D information, resulting in limited dynamics. In this work, instead of generating, we focus on identifying whether the generated video adheres to physical laws.

## B    DISCUSSION ON CATEGORIZATION

In this work, group various types of solids into a single, unified 'solid' category. In theory, all solids can be prescribed as a universal constitutive model governed by the conservation of mass and momentum. Nevertheless, graphics researchers typically model certain materials using simplified constitutive models to lower computational costs. For instance, a rigid body, in reality, is a deformable body with a very high stiffness. In fact, exact rigidity does not exist in the real world. Similarly, researchers propose other solid constitutive models (e.g. fabrics, granular materials) to simplify computation for some specific material behaviors. Our choice of solid, as a broad categorization, serves as a basis for generalization beyond specific simulation techniques used in graphics research. Our benchmark is designed to capture a wide spectrum of physically plausible behaviors, rather than isolating specific graphics sub-domains.

The interactions between solid-solid pairs primarily focus on resolving contact constraints to prevent penetration. In contrast, solid-fluid interactions exhibit more diversity. For example, water can be repelled by an umbrella but can be absorbed by a sponge. Such behaviors, including permeability, adhesion, and absorption, are not available in solid-solid interactions. Given the diversity of solid-fluid interactions, we choose to balance the sample counts for solid-solid and solid-fluid interactions to ensure our benchmark does not overemphasize simpler contact-based interactions while underrepresenting more complex and varied solid-fluid dynamics.

Our current design prioritizes simplicity and broad applicability, especially for non-experts who may use the benchmark across various disciplines. Future work can consider more fine-grained categorization to better measure the ability of generative models to handle specific properties (e.g., plasticity, viscosity) if desired.

## C    MORE DISCUSSION ON USING BINARY FEEDBACK

We highlight that binary feedback (0/1) is quite popular in aligning generative models such as large language models [31, 120]. Further, we observe that binary feedback is much easier to collect at industrial scale by big generative model providers (e.g., ChatGPT). For instance, we note that the ChatGPT user interface asks binary preference after generating the response to a simple query. Similar extensions exist in the field of text-to-image generative models [58, 54]. Hence, the binary feedback protocol is quite powerful in studying and improving the generative models.

We highlight that a dense feedback system would capture more nuanced mistakes of the video generative models (e.g., completing 8 movements versus 6 movements). However, designing such prompts is non-trivial, and evaluating the generated videos in such scenarios is much more challenging, labor-extensive, and expensive (especially with limited academic budgets). Further, we note that the collection of diverse and denser forms of feedback is a crucial future work.

In this work, we do not report posterior physical commonsense performance (PC = 1 given SA = 1) since they can be inferred from the joint and marginal scores. In addition, we note that a bad model can easily game the posterior metric. For example, a bad model can generate a video which aligns with the prompt for 1 out of 700 prompts in the dataset. Now, assume that this video is also accurate in terms of physical commonsense. Hence, the posterior performance of this model will be 100%. This can be quite misleading for the practitioners.

## D    LIMITATIONS

In this work, we evaluate the physical commonsense capabilities of T2V generative models. Specifically, we curated the VIDEOPHY dataset, consisting of 688 captions. We argue that the captions are comprehensive and high-quality after going through our three-stage data curation pipeline. In the future, it will be pertinent to expand the physical commonsense understanding to more branches of physics, including projective geometry. Also, as the recent trend of LLM [86, 63] begins to study the power of process reward modeling, we give up exploring more fine-grained evaluation signals like frame level alignment score of physics due to the heavy cost and leave it for future work. It can be interesting to study using fine-grained language feedback to update video generation model just like the reflection mechanism in LLM / VLM [96, 117, 116]. Additionally, we test a diverse

set of T2V generative models, including both open and closed models. While it is financially and computationally challenging to evaluate an exhaustive list of models, we have aimed to incorporate models with diverse architectures, training datasets, and inference strategies. In the future, it will be important to gain access to and include new high-performance T2V models in our study.

In addition, we perform human annotations using Amazon Mechanical Turkers (AMT), where most of the workers primarily belong to the US and Canada. Hence, the human annotations in this work do not represent the diverse demographics around the globe. As a result, our human annotations reflect the perceptual biases of the annotators from Western cultures. In the future, it will be pertinent to assess the impact of diverse groups on our human evaluations. Finally, we acknowledge that text-to-video generative models can perpetrate societal biases in their generated content [109, 4]. It is critical that future work quantifies this bias in the generated videos and provides methods for the safe deployment of the models. Also, since our video just covers the domain of physics-specific videos, it is interesting to see whether it can generalize and evaluate the videos with other properties such as long, egocentric, embodied related [44, 124, 1], etc.

## E    DATA LICENSING

The VIDEOPHY dataset comprises videos generated by various T2V (Text-to-Video) generative models, detailed in Section F. The licensing terms for these videos will align with those specified by the respective model owners, as cited in this work. The curated captions and human annotations will be licensed under the MIT License.

## F    VIDEO GENERATIVE MODELS

For the open models, we benchmark *Zeroscope* [21, 110], a latent diffusion-based text-to-video model that adapts the text-to-image generative model [92] for video generation by training on high-quality video and image data for enhanced visual quality. Further, we benchmark *LaVIE* [112], a cascaded video latent diffusion model instead of a single diffusion model. Specifically, the LaVIE model is trained with a specialized curated dataset for enhanced visual quality and diversity. In addition, we test *VideoCrafter2*, a latent diffusion T2V model that enhances video generation quality by training on high-quality image-text data [102]. In our study, we also benchmark *OpenSora* [80], an open-source effort to replicate Sora [18], a high-performant closed latent diffusion model that uses diffusion transformers [82] for text-to-video generation. Finally, we include *StableVideoDiffusion (SVD)* [13], a latent diffusion model that can generate high resolution videos conditioned on a text or image. Since SVD-I2V (Image-to-Video) is publicly available, we utilize that to generate the videos. Specifically, we utilize SD-XL-Base-1.0 [85] to generate the conditioning images from the captions in the VIDEOPHY dataset. We term the entire pipeline as *SVD-T2I2V*.

For the closed models, we include *Gen-2* [29], a closed latent video diffusion model from Runway. In addition, we include Pika [83] with undisclosed information about the underlying generative model. Specifically, we wrote a custom API to acquire Gen-2 and Pika videos after paying for their monthly subscription for a total of $225. Finally, we include two versions of the *Lumiere* [8] from Google research. Specifically, *Lumiere-T2V* generates a video conditioned on the text, while *Lumiere-T2I2V* generates a video conditioned on an image, that is in-turn generated with the caption using a text-to-image generative model [93]. CogVideoX [123] is a most recent open-sourced state-of-the-art video generation models, which uses a MMDiT [30]-like architecture, and achieves very good text-to-video alignment and video quality performance.

## G    QUERYING GPT-4 FOR PROMPT GENERATION

In this section we discuss the prompt we utilized to generate all the prompts including three physical interaction categories: solid-solid, solid-fluid, fluid-fluid for video generation, which is displayed in Table 6, Table 7 and Table 8.

Develop unique and imaginative captions, each briefly describing the interaction between two different solid materials in a realistic scene. Each caption should consist of 7-10 words and clearly indicate the solids involved in the action.

Guidelines:

1. Focus on common solids used in everyday scenarios, avoiding rare or seldom-used materials.

2. Exclude actions like 'celebrating', 'arguing', or 'laughing' that do not clearly involve physical interaction between materials.

3. Avoid generating static scenes (e.g., 'Lid covers pot to retain heat', 'Stack of paper sits on the desk').

4. Avoid adding participle phrases (e.g., 'sweetening it', 'a creamy swirl', 'fizzing energetically') in the caption.

5. The captions should focus on the actions that require contact forces, or friction forces. Do not focus on the actions that require penetration forces.

6. Format each caption as follows: 'action': ACTION, 'solid 1': SOLID, 'solid 2': SOLID, 'caption': CAPTION

Bad Examples Of Captions (Do Not Generate Such Captions):
A diamond scratching glass. ## Scratching action that requires penetration
A key scratches the surface of a wooden table. ## Scratching action that requires penetration

Good Examples Of Captions:
A brick presses down on a metal can.
A snowball falls to the ground and splits apart.
A small red elastic ball stuck to the wall.

Figure 6: GPT-4 Prompt to Generate Solid-Solid Captions.

Develop unique and imaginative captions, showcasing interaction between a solid material with a fluid material, for generating a video. After crafting the caption, list the entities that act as solid and fluid in the caption.

Guidelines:

1. Focus on common solids and fluids used in everyday scenarios, avoiding rare or seldom-used materials.

2. Exclude actions like 'celebrating', 'arguing', or 'laughing' that do not clearly involve physical interaction between materials.

3. Avoid actions that execute state change from solid to fluid or vice-versa.

4. Avoid generating static scenes (e.g., 'Lid covers pot to retain heat').

5. Avoid adding participle phrases (e.g., 'sweetening it', 'a creamy swirl', 'fizzing energetically') in the caption.

6. The captions should focus on the actions that require contact forces, or friction forces. Do not focus on the actions that require penetration forces.

7. Format each caption as follows: 'action': ACTION, 'solid': SOLID, 'fluid': FLUID, 'caption': CAPTION

Bad Examples Of Captions (Do Not Generate Such Captions):
Sugar dissolves in water. ## dissolving action will not be visible in video
Sulfuric acid corroding metal. ## corrosion will not be visible in video
Water boiling in a pot. ## boiling action will not be visible in video

Good Examples Of Captions:
A dam break releases a massive flood.
An iron rod falls into the water.
A metal spoon stirs the honey in a cup.

Figure 7: GPT-4 Prompt to Generate Solid-Fluid Captions.

---

Develop unique and imaginative captions, each briefly describing the interaction between two different fluid materials in a realistic scene. Each caption should consist of 7-10 words and clearly indicate the fluids involved in the action.

Guidelines:

1. Focus on common fluids used in everyday scenarios, avoiding rare or seldom-used materials.

2. Exclude actions like 'celebrating', 'arguing', or 'laughing' that do not clearly involve physical interaction between materials.

3. Avoid generating static scenes (e.g., 'Lid covers pot to retain heat').

4. Avoid adding participle phrases (e.g., 'sweetening it', 'a creamy swirl', 'fizzing energetically') in the caption.

5. The captions should focus on the actions that require mixing and laying for liquid-liquid interactions, or some contact forces between liquid and gas.

6. Format each caption as follows: 'action': ACTION, 'fluid 1': FLUID, 'fluid 2': FLUID, 'caption': CAPTION

Bad Examples Of Captions (Do Not Generate Such Captions):
Juice solidifies around water in ice trays. ## solidification won't be visible in the video
Sugar disappears into stirring water. ## dissolving won't be visible in the video An acid and a base react to neutralize each other, forming water. ## chemical reactions are not visible in the video

Good Examples Of Captions:
The wind creating ripples across the surface of the lake.
Milk falls into a transparent cup of water.
Oil falls into a transparent cup of water.

---

Figure 8: GPT-4 Prompt to Generate Fluid-Fluid Captions.

## H  HUMAN ANNOTATION SCREENSHOT

We display the screenshot of our human annotation system in Figure 9 .

Answer the following questions based on the caption and the generated video.

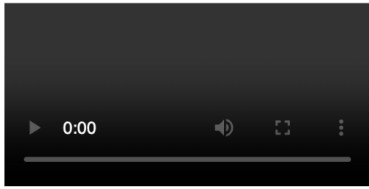

Caption: ${caption}

Does the video exhibit **Text Adherence** (Video-Text Alignment)?
○ Yes  ○ No

Does the video follow **Physics Laws or Physical Commonsense**? (This property is independent of Video-Text Alignment)
○ Yes  ○ No

**Submit**

Figure 9: The screenshot of the human annotation interface.

## I  VIDEOCON DETAILS

We prompt VIDEOCON to generate a text response (*Yes/No*) conditioned on the multimodal template $\mathcal{T}_t(x)$ for semantic adherence and physical commonsense tasks. Formally,

$$\mathcal{T}_t(x) = \begin{cases} \mathcal{T}_{SA}(V, C), & t = SA \\ \mathcal{T}_{PC}(V), & t = PC \end{cases} \tag{1}$$

where $t$ is either semantic adherence to the caption or physical commonsense task, $C$ is the conditioning caption and $V$ is the generated video for the caption $C$. We provide the multimodal templates

$(\mathcal{T}_{SA}(V,C), \mathcal{T}_{PC}(V))$. We compute the score from the VIDEOCON model $p_\theta$:

$$s_\theta(\mathcal{T}_t(x)) = \frac{p_\theta(Yes|\mathcal{T}_t(x))}{p_\theta(Yes|\mathcal{T}_t(x)) + p_\theta(No|\mathcal{T}_t(x))}, \qquad (2)$$

where $p_\theta(Yes|\mathcal{T}_t(x))$ is the probability of '*Yes*' conditioned on $\mathcal{T}_t(x)$, and $t \in \{SA, PC\}$. [8]

We present the prompts used for the GPT4V, Gemini-1.5-Pro-Vision, VideoCon baselines, and VIDEOCON-PHYSICS for semantic adherence evaluation in Figure 10 and physical commonsense alignment in Figure 11.

> **Semantic adherence**:
>
> **Given: V** (Video), **T** (Caption)
>
> **Instruction (I):** *[V] Does this video entail the description [T]?*
> **Response (R):** *Yes* or *No*

Figure 10: Template used assessing semantic adherence for a generated video.

> **Physical Commonsense**:
>
> **Given: V** (Video)
>
> **Instruction (I):** *[V] Does this video follow physical laws?*
> **Response (R):** *Yes* or *No*

Figure 11: Template for assessing physical commonsense. We note that the physical commonsense is independent of the conditioning caption. Hence, it is not present in this template.

## J  FINE-GRAINED DIVERSITY ANALYSIS

In this section, we visualize the fine-grained statistics of collections across different physical interaction categories (Figure 12 - Figure 14).

## K  RESULTS FOR TASK COMPLEXITY

We compare the performance of various video generative models across different task complexity in Table 6.

## L  FINE-GRAINED RESULTS

In this section, we report the fine-grained performance of semantic adherence and physical commonsense scores from all video generation models and compute the scores across different physical interaction categories (solid-solid, solid-fluid and fluid-fluid), as well as difficulty levels (0 and 1).

## M  AUTOMATIC EVALUATION BASELINES

Similar to [6], we utilize the capability of **GPT-4Vision** [79] to reason over multiple images in a zero-shot manner. Specifically, we prompt the GPT-4V model with the caption and 8 video frames

---

[8]As a large video multimodal model, VIDEOCON predicts a token distribution over the entire token vocabulary conditioned on the multimodal template. Therefore, $p_\theta(Yes|\mathcal{T}_t(x)) + p_\theta(No|\mathcal{T}_t(x))$ is not equal to 1.

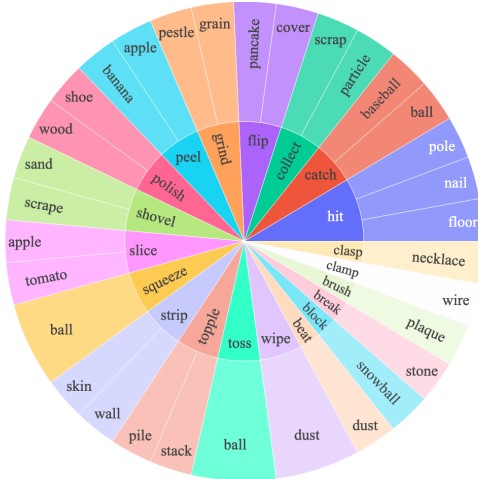

Figure 12: Top 20 most frequently occurring verbs (inner circle) and their top 4 direct nouns (outer circle) in our curated captions that consists of interaction between solid-solid states of matter.

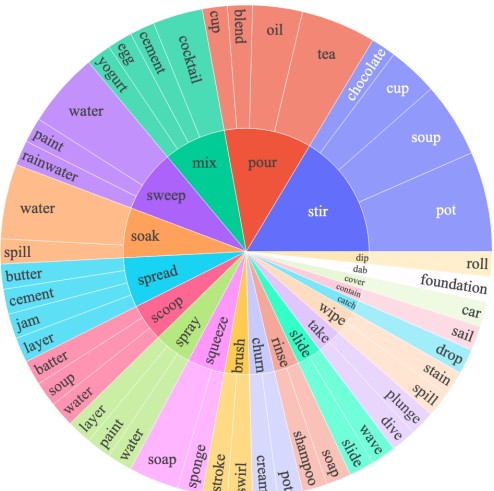

Figure 13: Top 20 most frequently occurring verbs (inner circle) and their top 4 direct nouns (outer circle) in our curated captions that consists of interaction between solid-fluid states of matter.

sampled uniformly from the generated video. Here, we instruct the model to provide the semantic adherence (0 or 1) and physical commonsense score (0 or 1). Since GPT-4V does not process videos natively, we assess the automatic evaluation using **Gemini-Pro-Vision-1.5**, which can input the caption and the entire generated video. Specifically, we instruct it to provide the semantic adherence (0 or 1) and physical commonsense (0 or 1) of the input video, identical to the GPT-4V analysis. We provide the prompts used in the experiments in Figure 10 and 11.

## N    INFERENCE DETAILS

We add the inference configurations for different video generation models in Table 9.

## O    TRAINING DETAILS FOR VIDEOCON-PHYSICS

To create VIDEOCON-PHYSICS, we use low-rank adaptation (LoRA) [39] of the VIDEOCON applied to all the layers of the attention blocks including QKVO, gate, up and down projection matrices. We set the LoRA $r = 32$ and $\alpha = 32$ and dropout = 0.05. The finetuning is performed for 5 epochs using

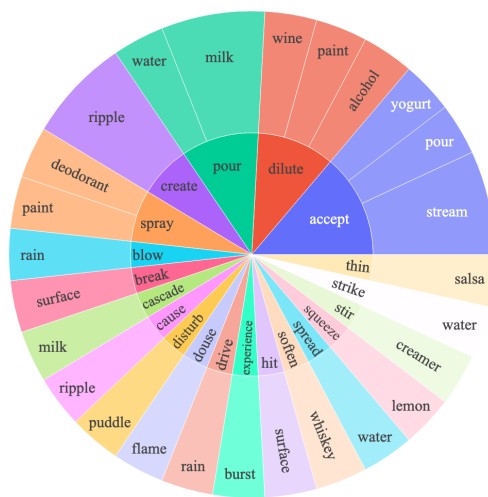

Figure 14: Top 20 most frequently occurring verbs (inner circle) and their top 4 direct nouns (outer circle) in curated captions that consists of interaction between fluid-fluid states of matter.

Table 6: **Fine-grained performance across caption complexity using human evaluation.** We find that T2V models struggle more on the harder captions than the easier captions in both the semantic adherence (SA) and physical commonsense (PC) metrics.

| Model | Easy (%) | | Hard (%) | |
|---|---|---|---|---|
| | SA | PC | SA | PC |
| *Open Models* | | | | |
| CogVideoX-5B | 63.8 | 55.3 | 62.5 | 50.3 |
| VideoCrafter2 | 53.4 | 38.1 | 42.6 | 30.3 |
| CogVideoX-2B | 51.1 | 38.3 | 42.6 | 29.0 |
| LaVIE | 51.9 | 31.2 | 44.8 | 24.0 |
| SVD-T2I2V | 41.8 | 37.6 | 43.2 | 22.6 |
| ZeroScope | 32.3 | 33.9 | 27.7 | 31.0 |
| OpenSora | 20.1 | 25.4 | 5.2 | 21.3 |
| *Closed Model* | | | | |
| Pika | 45.7 | 39.9 | 35.1 | 32.1 |
| Dream Machine | 65.2 | 29.4 | 57.8 | 12.5 |
| Lumiere-T2I2V | 56.6 | 29.1 | 38.7 | 20.0 |
| Lumiere-T2V | 38.6 | 34.9 | 38.1 | 19.4 |
| Gen-2 | 26.6 | 31.8 | 26.6 | 21.6 |

Adam [46] optimizer with a linear warmup of 50 steps followed by linear decay. Similar to [5], we chose the peak learning rate as $1e - 4$. We utilized 2 A6000 GPUs with the total batch size of 32. In addition, we finetune our model with 32 frames in the video and the frames are resized to $224 \times 224$ by image processor. Similar to [72, 125], we create 32 segments of the video, and sample the middle frame for each segment.

## P AUTOMATIC AND HUMAN LEADERBOARD

We compute the physical commonsense and semantic adherence scores for the models on the testing set using VIDEOCON-PHYSICS. Subsequently, we take their average and create a rankings of the models. We have a similar ranking of the models using the joint performance metrics (SA=1, PC=1) from human evaluation. We present the human and automatic leaderboard for the open and closed models in Table 10. Our analysis reveals that the average rank of the models in the automatic leaderboard is 0.66 above or below the expected rank of the model in the human leaderboard. This indicates VIDEOCON-PHYSICS is reliable for evaluating the future models on our dataset.

Table 7: Fine-grained performance of T2V models for the interaction between diverse states of matter using human evaluation. Ideally, we want the T2V models to achieve a high score on the *SA = 1 and PC = 1* metric while reduce the score on the *SA=0 and PC=0*, *SA=1 and PC=0*, and *SA=0 and PC=1* metrics.

| Source | Category | SA (%) | PC (%) | SA=1 and PC=1 (%) | SA=1 and PC=0 (%) | SA=0 and PC=1 (%) | SA=0 and PC=0 (%) |
|---|---|---|---|---|---|---|---|
| | | | | Open Models | | | |
| | Fluid-Fluid | 61.8 | 43.6 | 18.2 | 18.2 | 18.2 | 20.0 |
| CogVideoX-5B | Solid-Fluid | 76.6 | 59.3 | 53.1 | 23.4 | 6.2 | 17.2 |
| | Solid-Solid | 50.3 | 24.5 | 25.9 | 18.9 | 18.9 | 30.8 |
| | Fluid-Fluid | 34.5 | 47.3 | 25.5 | 9.1 | 21.8 | 43.6 |
| CogVideoX-2B | Solid-Fluid | 56.2 | 34.9 | 21.9 | 34.2 | 13.0 | 30.8 |
| | Solid-Solid | 43.0 | 28.2 | 12.7 | 30.3 | 15.5 | 41.5 |
| | Fluid-Fluid | 69.1 | 43.6 | 34.5 | 34.5 | 9.1 | 21.8 |
| LaVIE | Solid-Fluid | 52.1 | 30.8 | 15.8 | 36.3 | 15.1 | 32.9 |
| | Solid-Solid | 37.3 | 19.0 | 8.5 | 28.9 | 10.6 | 52.1 |
| | Fluid-Fluid | 12.7 | 27.3 | 7.3 | 5.5 | 20.0 | 67.3 |
| OpenSora | Solid-Fluid | 30.1 | 21.9 | 7.5 | 22.6 | 14.4 | 55.5 |
| | Solid-Solid | 7.7 | 23.8 | 1.4 | 6.3 | 22.4 | 69.9 |
| | Fluid-Fluid | 69.1 | 43.6 | 32.7 | 36.4 | 10.9 | 20.0 |
| VideoCrafter2 | Solid-Fluid | 57.5 | 41.8 | 27.4 | 30.1 | 14.4 | 28.1 |
| | Solid-Solid | 31.5 | 23.8 | 4.9 | 26.6 | 18.9 | 49.7 |
| | Fluid-Fluid | 58.2 | 34.5 | 18.2 | 40.0 | 16.4 | 25.5 |
| SVD-T2I2V | Solid-Fluid | 52.7 | 32.9 | 17.1 | 35.6 | 15.8 | 25.5 |
| | Solid-Solid | 25.9 | 27.3 | 4.2 | 21.7 | 23.1 | 51.0 |
| | Fluid-Fluid | 36.4 | 47.3 | 20.0 | 16.4 | 27.3 | 36.4 |
| ZeroScope | Solid-Fluid | 40.4 | 37.0 | 14.4 | 26.0 | 22.6 | 37.0 |
| | Solid-Solid | 17.5 | 22.4 | 6.3 | 11.2 | 16.1 | 66.4 |
| | | | | Closed Models | | | |
| | Fluid-Fluid | 76.4 | 10.9 | 9.1 | 67.3 | 1.8 | 21.8 |
| Dream Machine | Solid-Fluid | 68.1 | 23.6 | 16.7 | 51.4 | 6.9 | 25.0 |
| | Solid-Solid | 50.0 | 24.3 | 12.1 | 37.9 | 12.1 | 37.9 |
| | Fluid-Fluid | 37.7 | 26.4 | 15.1 | 22.6 | 11.3 | 50.9 |
| Gen-2 | Solid-Fluid | 38.5 | 18.5 | 8.1 | 30.4 | 10.4 | 51.1 |
| | Solid-Solid | 8.9 | 37.1 | 4.0 | 4.8 | 33.1 | 58.1 |
| | Fluid-Fluid | 45.4 | 34.5 | 9.1 | 36.4 | 25.5 | 29.1 |
| Lumiere-T2V | Solid-Fluid | 47.2 | 26.0 | 9.6 | 37.7 | 16.4 | 36.3 |
| | Solid-Solid | 26.5 | 27.3 | 8.4 | 18.2 | 18.9 | 54.5 |
| | Fluid-Fluid | 49.5 | 21.8 | 10.9 | 38.2 | 10.9 | 40.0 |
| Lumiere-T2I2V | Solid-Fluid | 59.6 | 26.0 | 17.1 | 42.5 | 8.9 | 31.5 |
| | Solid-Solid | 37.1 | 25.2 | 8.4 | 28.7 | 16.8 | 46.2 |
| | Fluid-Fluid | 68.0 | 58.0 | 44.0 | 24.0 | 14.0 | 18.0 |
| Pika | Solid-Fluid | 46.5 | 27.9 | 16.3 | 30.2 | 11.6 | 41.9 |
| | Solid-Solid | 24.8 | 36.8 | 13.6 | 11.2 | 23.2 | 52.0 |

Table 8: Fine-grained performance of T2V models for the complexity of the captions using human evaluation. Ideally, we want the T2V models to achieve a high score on the *SA = 1 and PC = 1* metric while reduce the score on the *SA=0 and PC=0*, *SA=1 and PC=0*, and *SA=0 and PC=1* metrics.

| Source | Category | SA (%) | PC (%) | SA=1 and PC=1 (%) | SA=1 and PC=0 (%) | SA=0 and PC=1 (%) | SA=0 and PC=0 (%) |
|---|---|---|---|---|---|---|---|
| | | | | Open Models | | | |
| CogVideoX-5B | EASY | 63.8 | 40.9 | 22.9 | 14.4 | 14.4 | 21.8 |
| | HARD | 62.6 | 38.1 | 24.5 | 12.3 | 12.3 | 25.2 |
| CogVideoX-2B | EASY | 51.1 | 38.3 | 20.7 | 17.6 | 17.6 | 31.4 |
| | HARD | 42.6 | 29.0 | 16.1 | 12.9 | 12.9 | 44.5 |
| LaVIE | EASY | 51.9 | 31.2 | 19.6 | 32.3 | 11.6 | 36.5 |
| | HARD | 44.8 | 24.0 | 11.0 | 33.8 | 13.0 | 42.2 |
| OpenSora | EASY | 20.1 | 25.4 | 4.8 | 15.3 | 20.6 | 59.3 |
| | HARD | 15.5 | 21.3 | 5.2 | 10.3 | 16.1 | 68.4 |
| VideoCrafter2 | EASY | 53.4 | 38.1 | 21.2 | 32.3 | 16.9 | 29.6 |
| | HARD | 42.6 | 30.3 | 16.1 | 26.5 | 14.2 | 43.2 |
| SVD-T2I2V | EASY | 42.0 | 38.0 | 16.0 | 25.0 | 21.0 | 37.0 |
| | HARD | 43.0 | 23.0 | 6.0 | 37.0 | 16.0 | 41.0 |
| ZeroScope | EASY | 32.3 | 33.9 | 13.8 | 18.5 | 20.1 | 47.6 |
| | HARD | 27.7 | 31.0 | 9.7 | 18.1 | 21.3 | 51.0 |
| | | | | Closed Models | | | |
| Dream Machine | EASY | 65.2 | 29.4 | 19.8 | 45.5 | 9.6 | 25.1 |
| | HARD | 57.9 | 12.5 | 5.9 | 52.0 | 6.6 | 35.5 |
| Gen-2 | EASY | 26.6 | 31.8 | 10.4 | 16.2 | 21.4 | 52.0 |
| | HARD | 26.6 | 21.6 | 4.3 | 22.3 | 17.3 | 56.1 |
| Lumiere-T2V | EASY | 38.6 | 34.9 | 11.1 | 27.5 | 23.8 | 37.6 |
| | HARD | 38.1 | 19.3 | 6.5 | 31.6 | 12.9 | 49.0 |
| Lumiere-T2I2V | EASY | 56.6 | 29.1 | 16.4 | 40.2 | 12.7 | 30.7 |
| | HARD | 38.7 | 20.0 | 7.7 | 31.0 | 12.3 | 49.0 |
| Pika | EASY | 45.7 | 39.9 | 23.7 | 22.0 | 16.2 | 38.2 |
| | HARD | 35.1 | 32.1 | 14.5 | 20.6 | 17.6 | 47.3 |

Table 9: Inference details for models in our testbed. Here, NA indicates that the information is not available for the closed models.

| Model | Resolution | # of Video Frames | Guidance Scale | Sampling Steps | Noise Scheduler |
|---|---|---|---|---|---|
| *Open Models* | | | | | |
| CogVideoX | 480 × 720 | 25 | 7.5 | 50 | DDPM [37] |
| ZeroScope | 320 × 576 | 32 | 9 | 50 | DPMSolverMultiStep [70] |
| VideoCrafter2 | 320 × 512 | 32 | 12 | 50 | DDIM [100] |
| LaVIE | 320 × 512 | 32 | 7.5 | 50 | DDPM [37] |
| OpenSora | 240 × 426 | 32 | 7 | 100 | IDDPM [76] |
| SVD-T2I2V | 1024 × 576 | 25 | (1, 3) | 25 | EulerDiscrete [43] |
| *Closed Models* | | | | | |
| Lumiere-T2V | 1024 × 1024 | 80 | 8 | 256 | NA |
| Lumiere-T2I2V | 1024 × 1024 | 80 | 6 | 256 | NA |
| Gen-2 | 720 × 1280 | 32 | 8.5 | 100 | NA |
| Dream Machine | 1280 × 720 | 24 | NA | NA | NA |
| Pika | 640 × 1088 | 72 | 12 | NA | NA |

# Q   ADDITIONAL EXPERIMENT: RELATIVE RANKINGS

In this work, we focus on collecting the absolute (0/1) feedback from the human annotators. Here, we aim to understand the effect of changing the feedback acquisition protocol to relative rankings for physical commonsense evaluation. Specifically, we ask the three workers to look at two videos simultaneously and pick the one with better physical commonsense. In particular, we got 500 pairwise comparisons for 4 video generative models (CogVideoX-5B, Pika, Gen2, OpenSora). It costs us $360 to run this human evaluation. Subsequently, we computed the ELO scores of these models based on the human annotations. We present the results in Table 12.

Interestingly, we find that the relative ranking of these models remains unchanged under both the feedback methods. Specifically, CogVideoX-5B and OpenSora are still the best and worst models on the VideoPhy dataset, respectively. We note that the open (usually smaller) video generative models will be penalized for losing to close (usually larger) video generative models in the ranking-based

Table 10: **Human and Automatic leaderboard for open and closed video generative models.** We compute the joint performance metrics (SA = 1, PC = 1) from human evaluation, and average the SA and PC scores from automatic evaluation to construct the leaderboard. The models are ranked from best to worst (descending order). We find that the automatic leaderboard reliably tracks the human leaderboard.

| Open models | | | Closed models | | |
|---|---|---|---|---|---|
| **Human** | **VIDEOCON-PHYSICS** | **Rank diff.** | **Human** | **VIDEOCON-PHYSICS** | **Rank diff.** |
| CogVideoX-5B | CogVideoX-5B | 0 | Pika | Dream Machine | 3 |
| VideoCrafter2 | VideoCrafter2 | 0 | Dream Machine | Lumiere-T2I2V | 1 |
| CogVideoX-2B | LaVIE | 1 | Lumiere-T2I2V | SVD-T2I2V | 1 |
| LaVIE | CogVideoX-2B | 1 | Lumiere-T2V | Pika | 1 |
| SVD-T2I2V | SVD-T2I2V | 0 | Gen-2 | Gen-2 | 0 |
| ZeroScope | ZeroScope | 0 | - | - | - |
| OpenSora | OpenSora | 0 | - | - | - |

Table 11: **Automatic Leaderboard on Video Generation Models.**

| # | Open Models | PC | SA | Avg. |
|---|---|---|---|---|
| 1 | CogVideo-xl | 41 | 57 | 49 |
| 2 | VideoCrafter-2 | 36 | 47 | 42 |
| 3 | LaVIE | 35 | 46 | 41 |
| 4 | Mochi | 30 | 42 | 36 |
| 5 | CogVideo (Base) | 29 | 40 | 35 |
| 6 | SDXV-T2i2V | 28 | 38 | 33 |
| 7 | Hunyuan Video | 26 | 38 | 32 |
| 8 | ZeroScope | 24 | 35 | 29 |
| 9 | Pika | 23 | 34 | 29 |

| # | Closed Models | PC | SA | Avg. |
|---|---|---|---|---|
| 1 | Luma Dreamer T2i2V | 30 | 45 | 38 |
| 2 | Lamma T2i2V | 25 | 36 | 31 |
| 3 | Pika | 23 | 34 | 29 |
| 4 | Gen-2 (Runway) | 31 | 26 | 29 |

setup. The absolute feedback operates independently across all video generative models, and helps in better contextualizing the capability of the models with similar scales.

# R  APPLICATIONS OF VIDEOCON-PHYSICS

In this work, we propose VIDEOCON-PHYSICS, an auto-evaluator that judges the semantic adherence and physical commonsense of the generated videos for a given caption. Here, we describe the potential usecases of the model for future work.

**Video Generative Model Selection:**    The ability to perform model verification on downstream tasks cheaply and reliably is critical. In this regard, the model builders can utilize VIDEOCON-PHYSICS to evaluate their candidate models on the VIDEOPHY dataset at scale. The top candidate models can then be evaluated with the human workers for more accurate evaluation.

**Data Filtering:**    With the advent of foundation models that are trained on the internet data, high-quality filtering has emerged as a crucial step in the pipeline [33, 131, 62]. Here, the data builders can utilize VIDEOCON-PHYSICS to filter low-quality video-text data that lacks in semantic adherence and physical commonsense.

**Post-training:**    Recently, aligning the generative models with human or AI feedback has become pivotal for high-quality generations [94, 89, 7, 108, 58]. Here, the post-training pipeline of the video generative models can leverage the VIDEOCON-PHYSICS model as an reward model that provides feedback to the model generated content. Subsequently, this feedback can be utilized to refine the model for better generations.

# S  CORRELATION WITH VIDEO QUALITY AND MOTION

There are several works that focus on assessing the generated video quality and motions [42, 67]. Here, we aim to assess the correlation between the semantic adherence and physical commonsense

Table 12: **Results with relative feedback acquisition protocol.** We present the ELO score of a few selected model by asking human annotators to compare the two videos side-by-side, and pick the one that follows physical commonsense more. We also mention that the binary ratings that the videos get under the absolute feedback acquisition protocol. We find that the rankings of the models are identical under both the setups.

| Model | ELO | Binary ratings |
|---|---|---|
| CogVideoX-5B | 1081 | 53 |
| Pika | 1048 | 36.5 |
| Gen2 | 1010 | 27.2 |
| OpenSora | 860 | 23.5 |

scores and these metrics. Specifically, we calculate the video quality using LAION aesthetic classifier [52] and video motion using the RAFT optical flow model [104]. Subsequently, we calculate the pearson correlation between video quality and motion with physical commonsense and semantic adherence. We present the results in Table 13.

We find that physical commonsense and semantic adherence are correlated positively with the video quality, albeit the correlation is not very strong. In addition, we find that physical commonsense and semantic adherence are negatively correlated with the video motions. This indicates that the video models tend to make more mistakes in the semantic adherence and physical commonsense when more motions are depicted in them. In this work, we consider a wide breadth of video generative models – open and closed. The closed models (Dream Machine/Gen-2/Pika) contribute to the higher end of the video quality while open models (Zeroscope/OpenSora) contribute to the lower end. While the high quality is 'correlated' with the better physical commonsense, we note that the absolute performance of the models is quite poor on our benchmark. For instance, Gen-2 achieves the one of the highest video quality score (5.8 on LAION aesthetics classifier) but has a poor semantic adherence and physical commonsense score of 7.6 (Table 3).

Table 13: **Correlation between video quality (Aesthetics) and optical flow (motion) with physical commonsense and semantic adherence over VIDEOPHY dataset.**

| Metrics | Correlation |
|---|---|
| Aesthetics-Physical Commonsense | 0.3 |
| Aesthetics-Semantic Adherence | 0.5 |
| Motion-Physical Commonsense | -0.8 |
| Motion-Semantic Adherence | -0.1 |

## T    FINETUNING VIDEO MODEL WITH VIDEOPHY DATA

This work is centered around physical commonsense evaluation, and we trained an automatic evaluator (VIDEOCON-PHYSICS) using the training set. Here, we assess whether VIDEOPHY training set instances can also be used for finetuning video models. Specifically, we finetune Lumiere-T2I2V model on the instances from the training set of VIDEOPHY which achieve a score of 1 on physical commonsense and a score of 1 on semantic adherence. In total, there are 1000 such (video, caption) pairs in the dataset. Post-finetuning, we generate the videos for the test prompts and evaluate them using our automatic evaluator, VIDEOCON-PHYSICS.

Table 14: **Finetuning Lumiere-T2I2V with the (video, text) pairs that achieve joint performance score of 1 (i.e., PC = 1 and SA = 1) in the train set of VIDEOPHY data.** While the training set of the VIDEOPHY was primarily collected for training an automatic evaluator, we test whether it can also improve the video generative models itself.

| Model | SA | PC | Average |
|---|---|---|---|
| Lumiere-T2I2V-Pretrained | 46 | 25 | 35 |
| Lumiere-T2I2V-Finetuned | 36.5 | 24.6 | 30.5 |

We present the results in Table 14. We find that the semantic adherence (video-text alignment) reduces by a large margin and physical commonsense remains unchanged after finetuning. This can be due to several factors: (a) the number of training samples is not enough, (b) optimization difficulties since the training videos are generated from several generative models (mix of on-policy and off-policy videos), and (c) vanilla finetuning being a bad algorithm for learning from these samples. Since post-training of video generative models is a less explored direction, there can be many ways to improve the generative model's physical commonsense. These results also show that mere finetuning with the samples in the training set of VideoPhy does not lead to large gains in the automatic evaluation on the test set. Future work will focus on training better physical commonsense models using the insights provided in our work.

## U    More Qualitative Examples of Poor Physical Commonsense

We present more examples from each generative model where one or more physical laws are violated in Figure 15 - Figure 26.

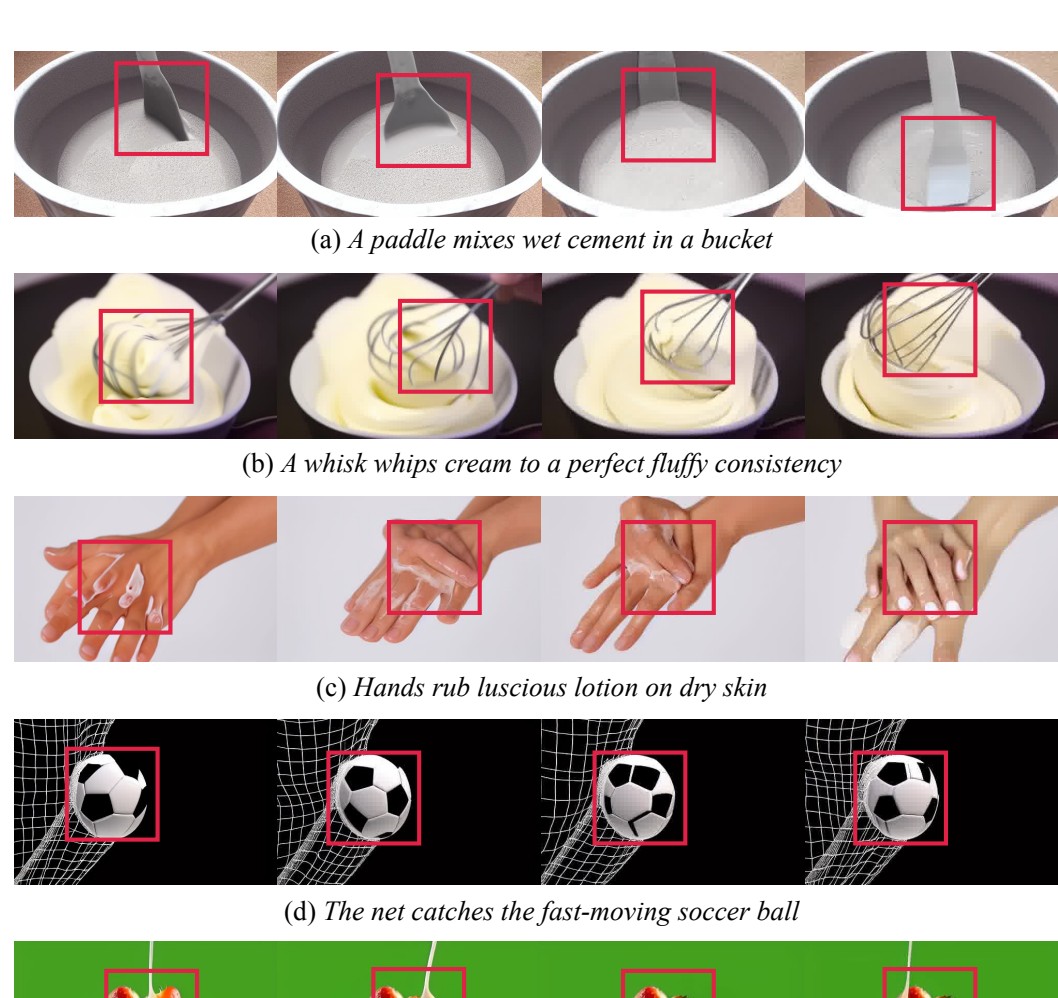

(a) *A paddle mixes wet cement in a bucket*

(b) *A whisk whips cream to a perfect fluffy consistency*

(c) *Hands rub luscious lotion on dry skin*

(d) *The net catches the fast-moving soccer ball*

(e) *Yogurt merging with strawberry puree*

Figure 15: Unphysical Generated Examples of LaVIE. (a) Solid Constitutive Laws Violation: the metal spoon should not deform; Nonphysical Penetration: the spoon unnaturally passes through the liquid. (b) Solid Constitutive Laws Violation: the whisk exhibits abnormal shape deformation. (c) Solid Constitutive Laws Violation: the two hands show abnormal shape deformation; Nonphysical Penetration: fingers penetrate each other; Conservation of Mass Violation: the geometry (plus texture) of the two hands are inconsistent over time. (d) Conservation of Mass Violation: the geometry (plus texture) of the soccer is inconsistent over time; Newton's Second Law Violation: the soccer does not fall under gravity. (e) Conservation of Mass Violation: the volume of yogurt in the cup does not increase as more yogurt is added.

## V EXAMPLES FROM DIVERSE STATES OF MATTER AND COMPLEXITY

We present a few qualitative examples highlighting instances of good physical commonsense and bad physical commonsense in Figure 27-Figure 29.

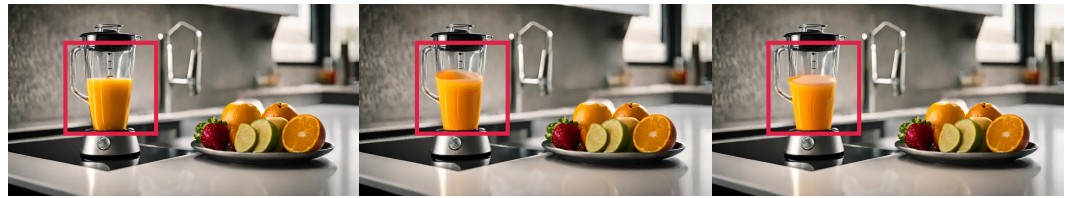

(a) *A blender spins, mixing squeezed juice within it*

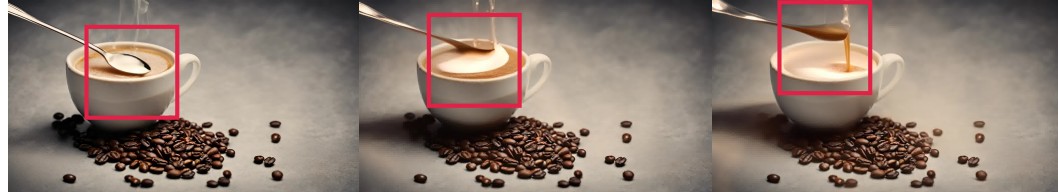

(b) *A teaspoon stirs sugar into a cup of coffee*

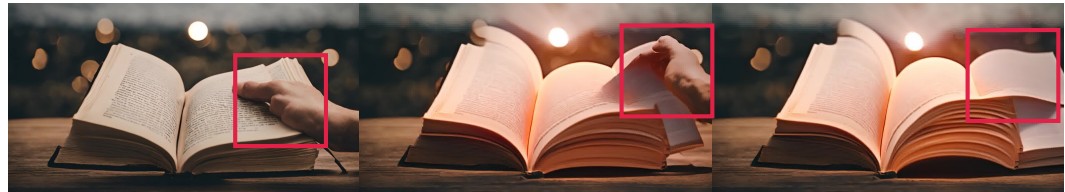

(c) *Hand flipping open book cover*

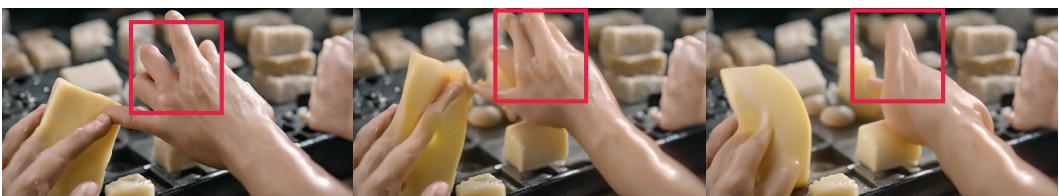

(d) *Soap washes grime off dirty hands*

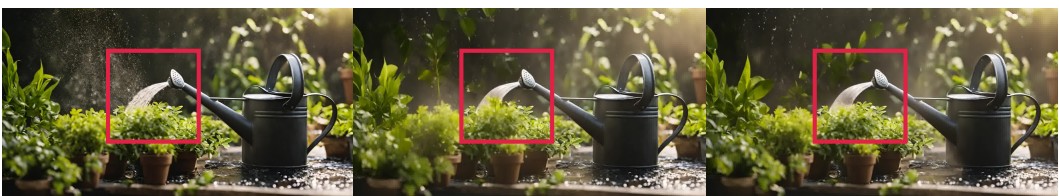

(e) *Water pouring from a watering can onto plants*

Figure 16: Unphysical Generated Examples of Gen-2. (a) Conservation of Mass Violation: the volume of juice in the blender increases over time without new substances being added. (b) Solid Constitutive Laws Violation: the metal spoon should not deform. (c) Conservation of Mass Violation: the volume of the book increases over time; Nonphysical Penetration: the fingers pass through the book. (d) Nonphysical Penetration: fingers penetrate into each other. (e) Newton's Second Law Violation: the flowing water appears to be static, ignoring the effect of gravity.

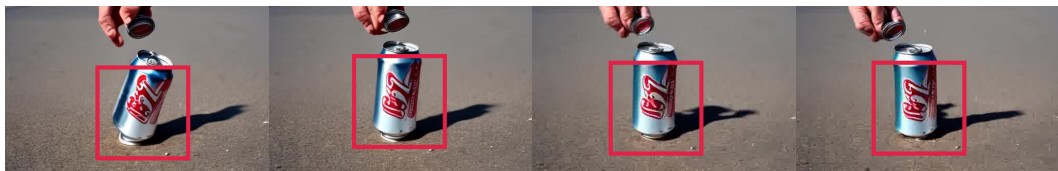

(a) *A foot crushing an empty soda can*

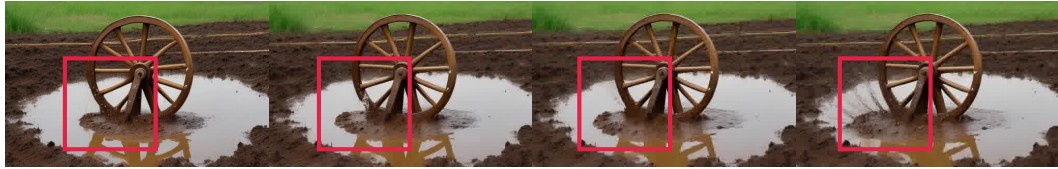

(b) *A spinning wheel sprays muddy water*

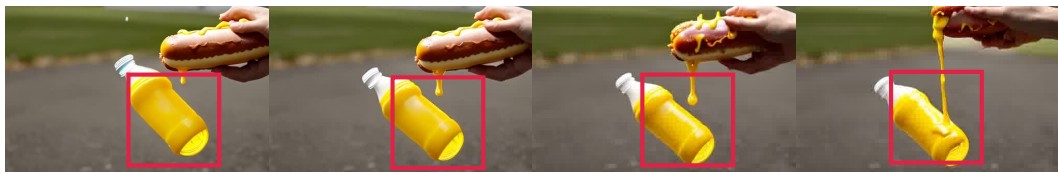

(c) *Mustard squirting out of a plastic bottle onto a hotdog*

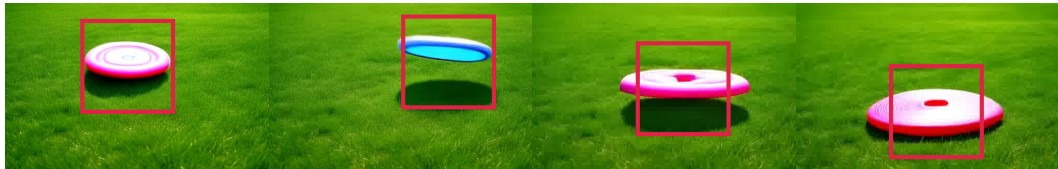

(d) *Plastic frisbee lands on a lush grass lawn*

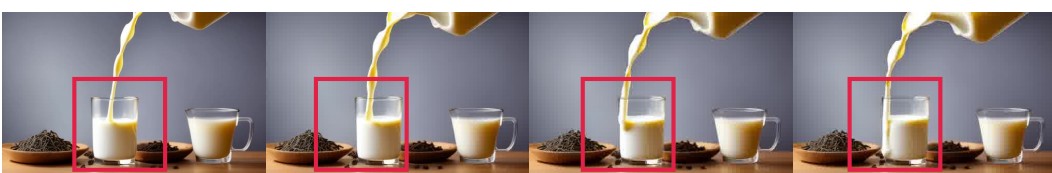

(e) *Pouring milk into still tea*

Figure 17: Unphysical Generated Examples of VideoCrafter2. (a) Newton's Second Law Violation: the metal can deforms without being pressed. (b) Newton's Second Law Violation: Water splashes while the rolling wheel remains static. (c) Newton's Second Law Violation: the bottle floats in the air, ignoring the effect of gravity; Fluid Constitutive Law Violation: the dripping and flowing of mustard are unnatural. (d) Conservation of Mass Violation: the geometry (plus texture) of the frisbee is not consistent over time. (e) Conservation of Mass Violation: the total volume of milk in the glass does not increase as more milk is poured into; Nonphysical Penetration: milk penetrates the glass.

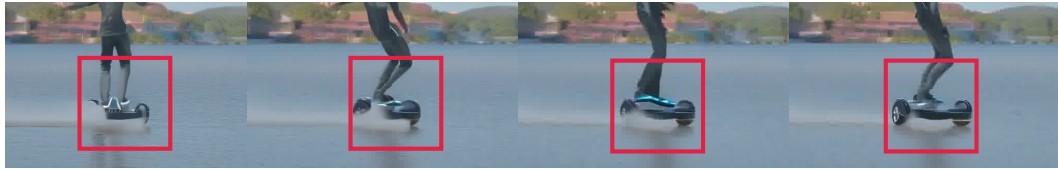

(a) *A futuristic hoverboard hovers just above the water*

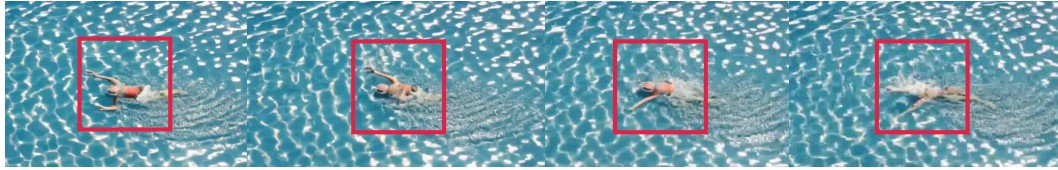

(b) *A swimmer splashing in the sea water*

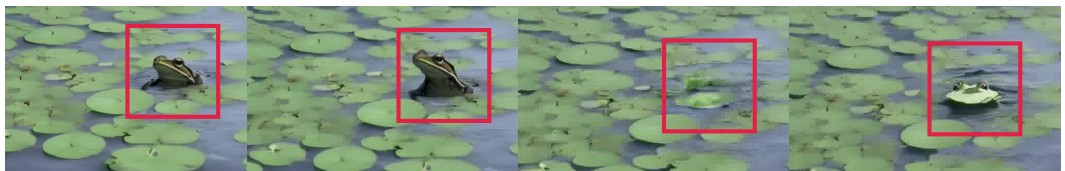

(c) *Frog leaping from one lilypad to another*

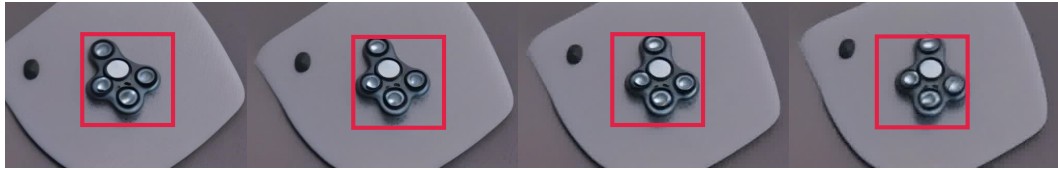

(d) *Plastic fidget spinner rotating on rubber mat*



(e) *The eraser rubs against the paper, removing pencil marks*

Figure 18: Unphysical Generated Examples of ZeroScope. (a) Newton's Second Law Violation: the motion of the hoverboard does not satisfy the momentum equation. (b) Newton's Second Law Violation: the motion of the arm of the swimmer is unnatural. (c) Conservation of Mass Violation: the geometry (texture) of the frog is inconsistent over time. (d) Newton's First Law Violation: the velocity of the fidget spinner changes despite being in a balanced state. (e) Solid Constitutive Laws Violation: the paper is torn apart without external forces but recovers later.

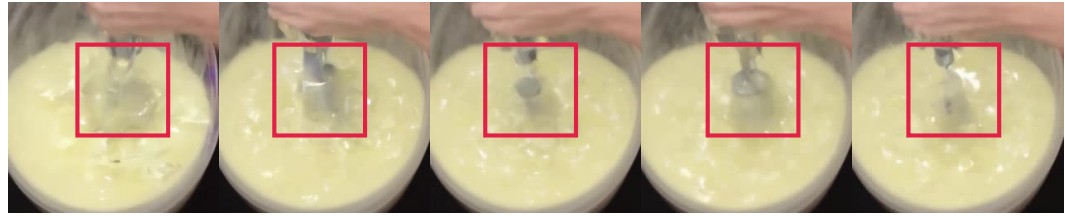

(a) *A blender spins, mixing squeezed juice within it*

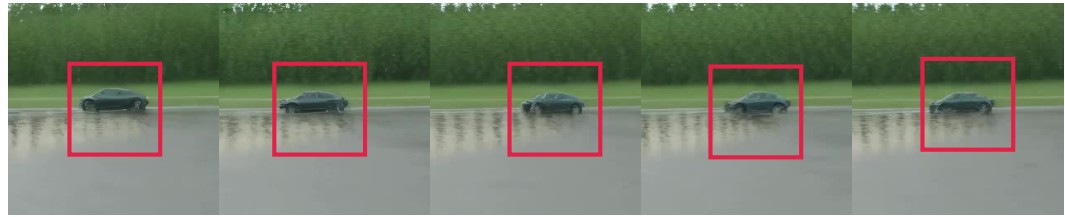

(b) *A car gliding over a road slick with rainwater*

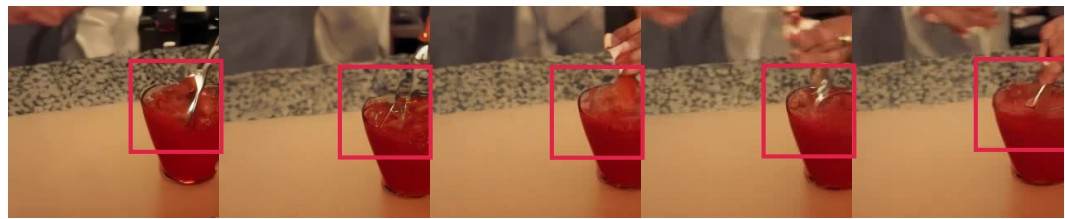

(c) *A shaker mixes a delightful cocktail at the bar*

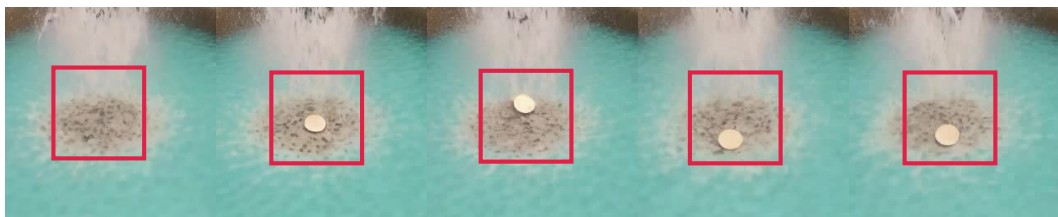

(d) *A shiny coin takes a dive into a clear water fountain*

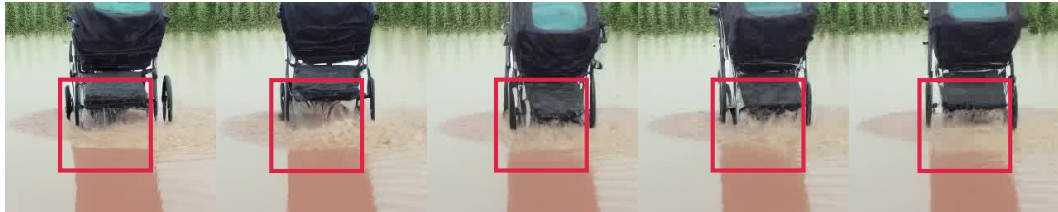

(e) *A stroller wheels through a large puddle*

Figure 19: Unphysical Generated Examples of OpenSora. (a) Solid Constitutive Laws Violation: the metal blender should not deform. (b) Newton's Second Law Violation: the car moves backward, violating the momentum equation (c) Solid Constitutive Laws Violation: the metal spoon deforms when stirring the cocktail. (d) Newton's First Law Violation: the coin moves on the ground back and forth without horizontal forces. (e) Conservation of Mass Violation: the left rear wheel disappears over time.

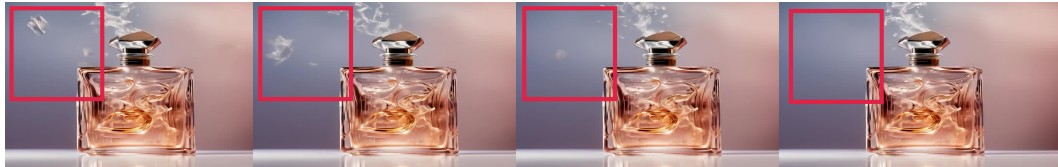

(a) *A perfume bottle spritzes fragrance into the air*

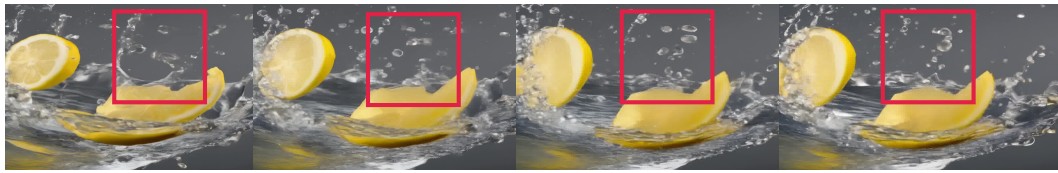

(b) *Lemon juice drops splash into water*

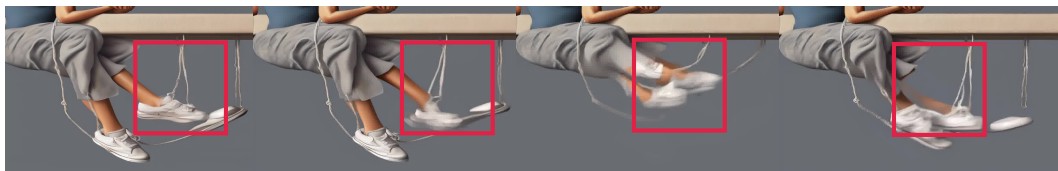

(c) *Loose sneaker swings on dangling foot*

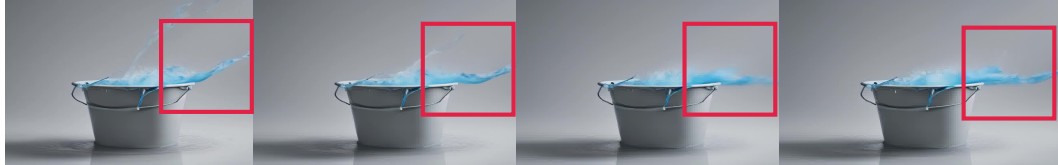

(d) *Detergent flowing into a bucket of water*

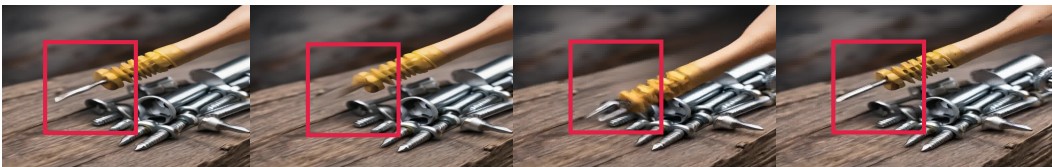

(e) *The screwdriver tightens the metal screw in the wood*

Figure 20: Unphysical Generated Examples of SVD-T2I2V. (a) Newton's Second Law Violation: the perfume spreads back and forth, violating the momentum equation. (b) Newton's Second Law Violation: the water drops float in the air, ignoring gravity. (c) Solid Constitutive Laws Violation: the leg exhibits unnatural deformation. (d) Newton's Second Law Violation: the water flows upward into the air without external forces. (e) Solid Constitutive Laws Violation: the screwdriver head deforms unnaturally.

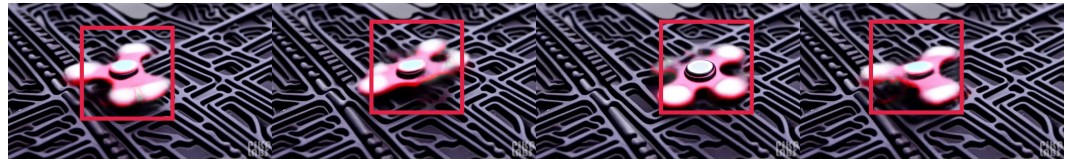

(a) *Plastic fidget spinner rotating on rubber mat*

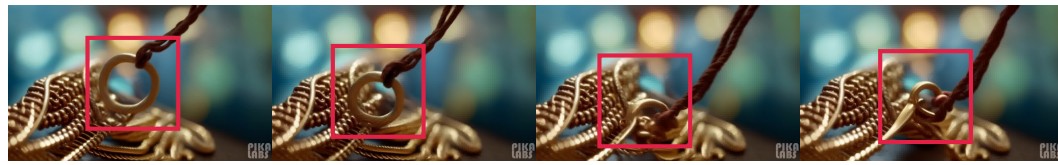

(b) *Clasping a necklace around a neck*

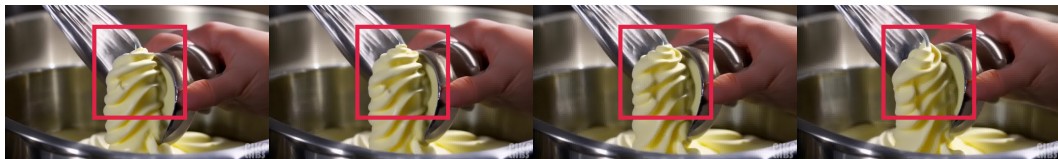

(c) *A whisk churns heavy cream into whipped cream*

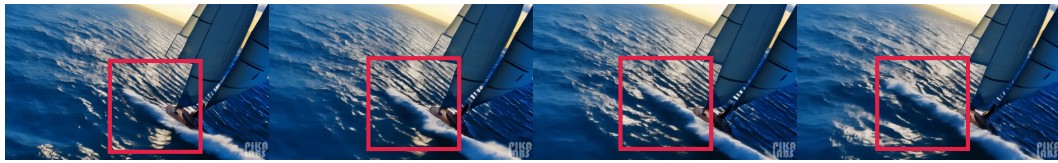

(d) *A sailboat cuts through the choppy sea waves*

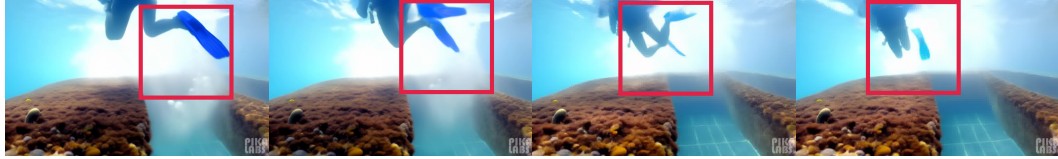

(e) *A diver plunges headlong into a sparkling pool*

Figure 21: Unphysical Generated Examples of Pika. (a) Solid Constitutive Laws Violation: the fidget spinner should not deform. (b) Solid Constitutive Laws Violation: the necklace should not deform. (c) Conservation of Mass Violation: the volume of cream increases over time without additional input. (d) Fluid constitutive Law Violation: unnatural waves on the sea surface. (e) Solid Constitutive Law Violation: one diving shoe splits into two and detaches from the feet.

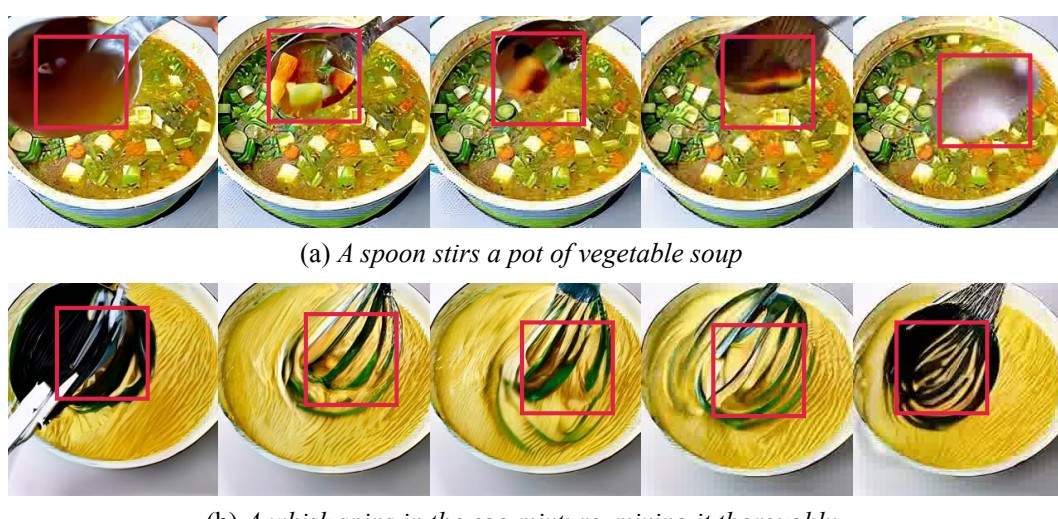

(a) *A spoon stirs a pot of vegetable soup*

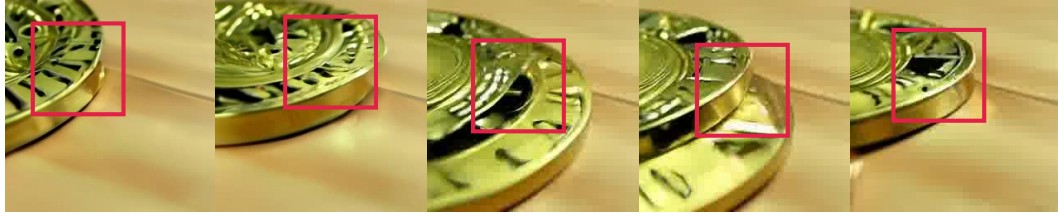

(b) *A whisk spins in the egg mixture, mixing it thoroughly*

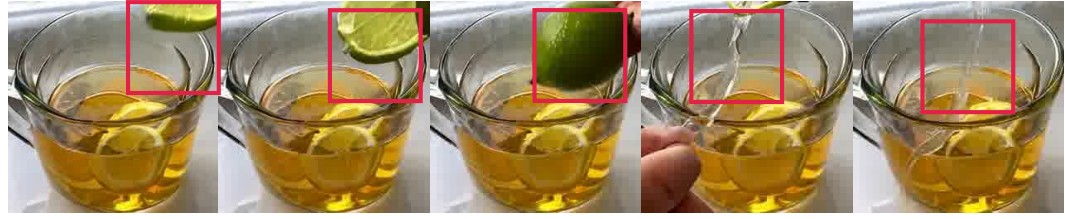

(c) *Coin spins rapidly on a wooden table*

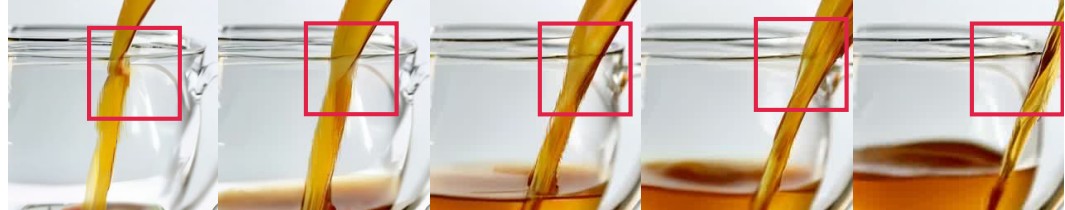

(d) *Squeezing lemon drops into warm tea*

(e) *Tea accepts stream of milk*

Figure 22: Unphysical Generated Examples of Lumiere-T2V. (a) Conservation of Mass Violation: the vegetable appears on the spoon out of nowhere. (b) Solid Constitutive Laws Violation: the whisk should not deform. (c) Solid Constitutive Laws Violation: the coin splits into two and then merges back into one. (d) Solid Constitutive Laws Violation: the lemon shows an unnatural appearance change; Fluid Constitutive Laws Violation: the lemon juice appears like static glue. (e) Nonphysical Penetration: the tea flows through the cup.

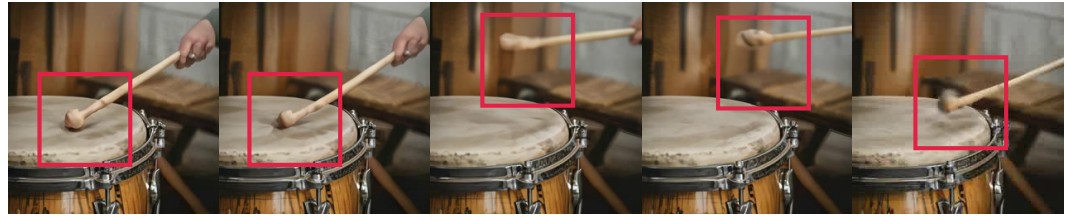

(a) *A drum vibrating from the beating stick*

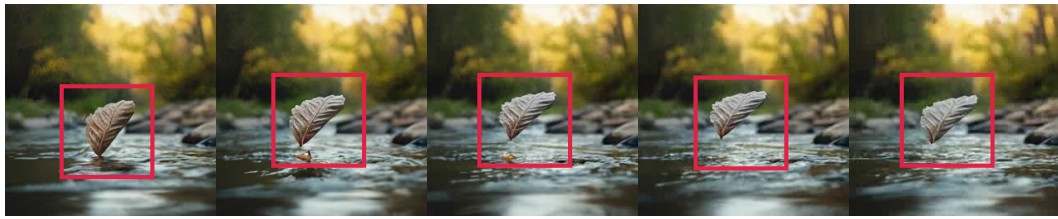

(b) *A leaf falls delicately into a slow-moving river*

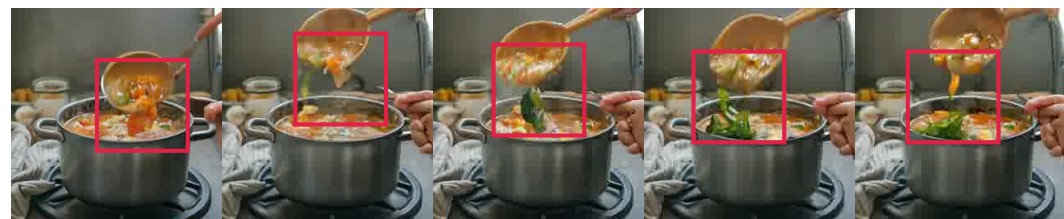

(c) *A wooden spoon stirring soup in a pot*

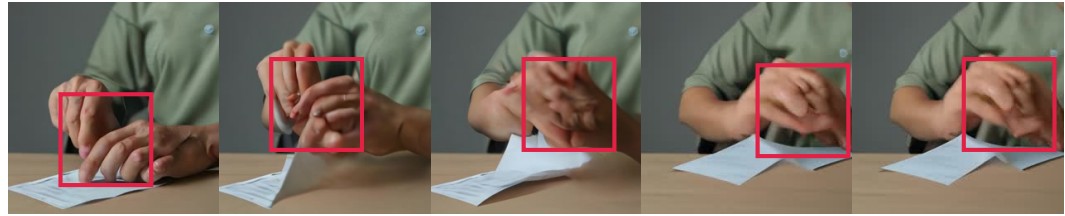

(d) *Hand folds the paper*

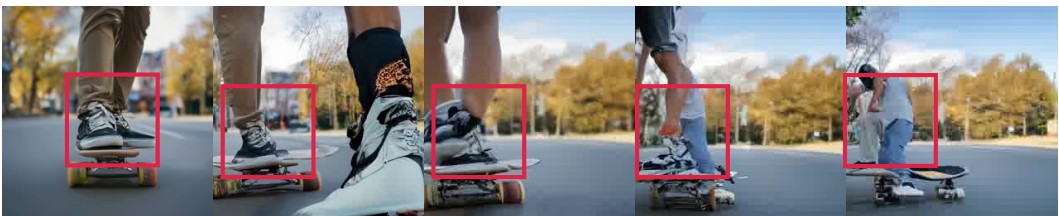

(e) *Skateboard glides on the pavement*

Figure 23: Unphysical Generated Examples of Lumiere-T2I2V. (a) Solid Constitutive Laws Violation: the drum stick head should not deform (b) Newton's Second Law Violation: the leaf floats in the air, ignoring gravity. (c) Conservation of Mass Violation: the vegetable appears on the spoon out of nowhere. (d) Nonphysical Penetration: hands penetrate each other. (e) Solid Constitutive Laws Violation: one leg on the skateboard transforms into a person.

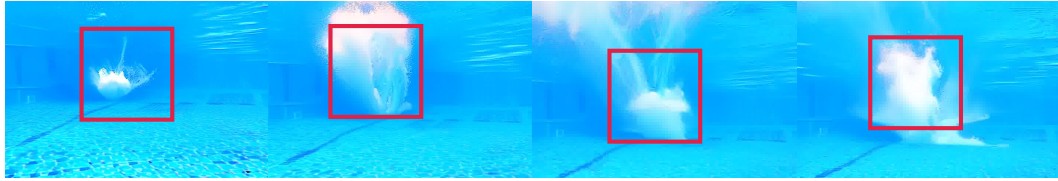

(a) *A diver splashing into the pool water*

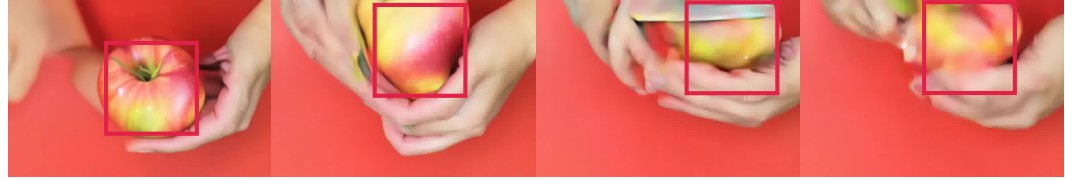

(b) *Peeler peels an apple*

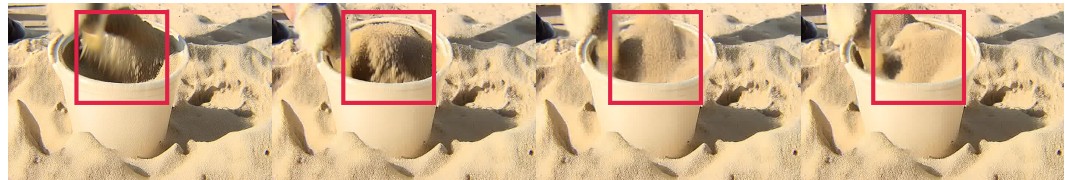

(c) *Shoveling sand into a bucket*

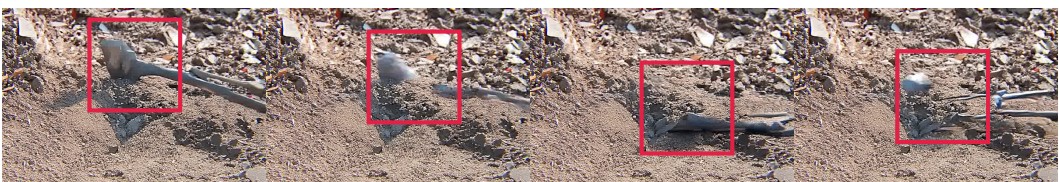

(d) *The pickaxe digs into the hard grounds*

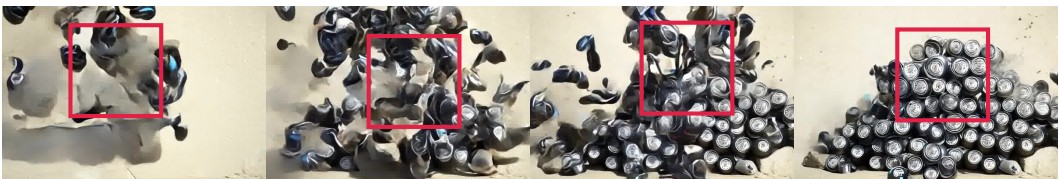

(e) *The rock topples the carefully stacked pile of cans*

Figure 24: Unphysical Generated Examples of CogVideoX-2B. a) Newton's Second Law Violation: the splash appears without any external force, violating the principle of momentum conservation. (b) Solid Constitutive Law Violation: the apple undergoes deformation, which should not occur. (c) Conservation of Mass Violation: the sand's volume changes over time without the addition of new material. (d) Conservation of Mass Violation: the geometry and texture of the pickaxe change inconsistently over time. (e) Solid Constitutive Law Violation: the cans exhibit unnatural deformations.

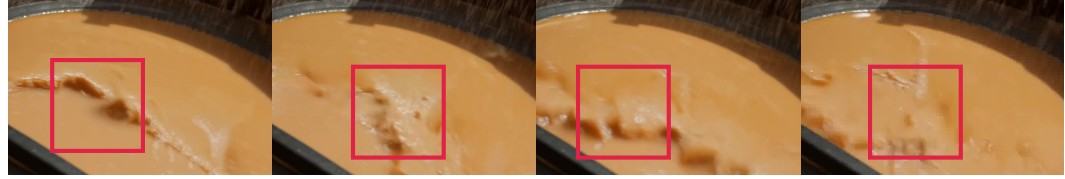

(a) *A spinning wheel sprays muddy water*

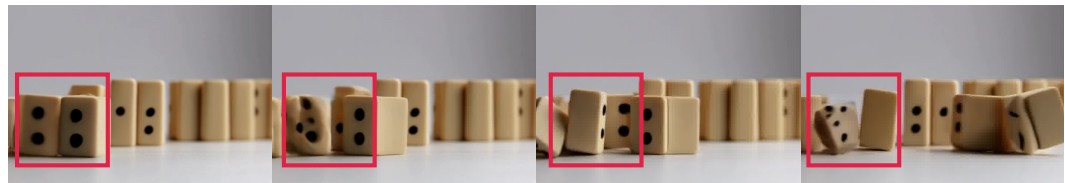

(b) *Dominoes toppling one after another on the table*

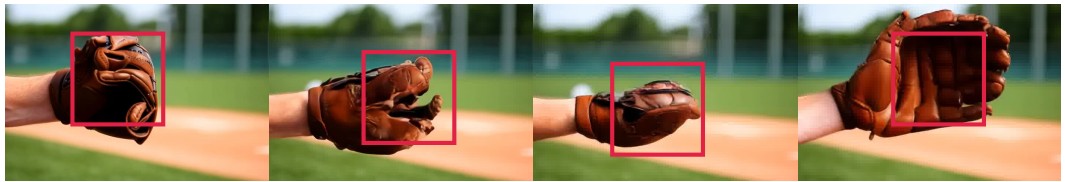

(c) *Leather glove catching a hard baseball*

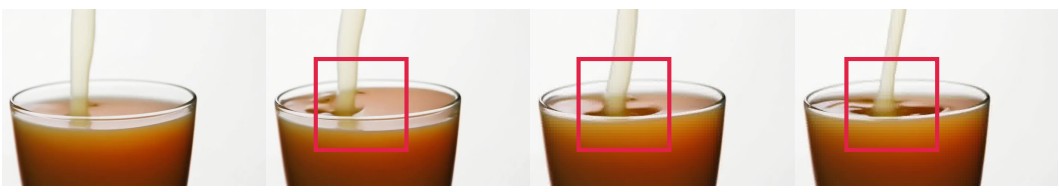

(d) *Milk blending seamlessly into tea*

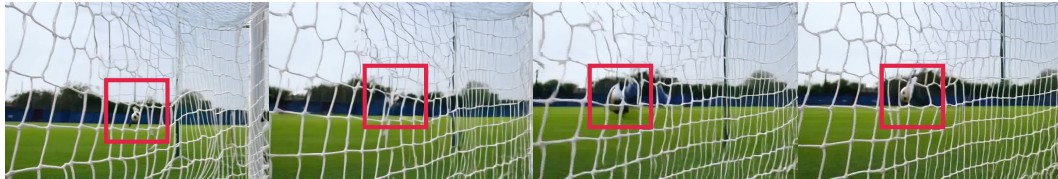

(e) *The net catches the fast-moving soccer ball*

Figure 25: Unphysical Generated Examples of CogVideoX-5B. (a) Newton's Second Law Violation: the wave dynamics are discontinuous over time, violating the momentum conservation principle. (b) Conservation of Mass Violation: the geometry and texture of the dominoes change inconsistently over time. (c) Solid Constitutive Law Violation: the leather glove exhibits unnatural deformations. (d) Conservation of Mass Violation: the volume of tea remains unchanged despite the addition of milk to the glass cup. (e) Solid Constitutive Law Violation: the soccer ball displays unnatural and discontinuous deformations over time.

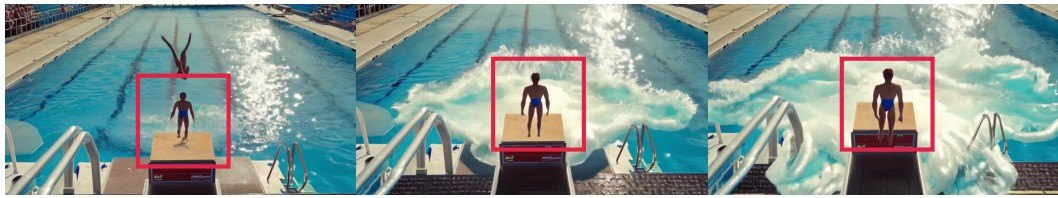

(a) *A brave diver splashes into a pool from a great height*

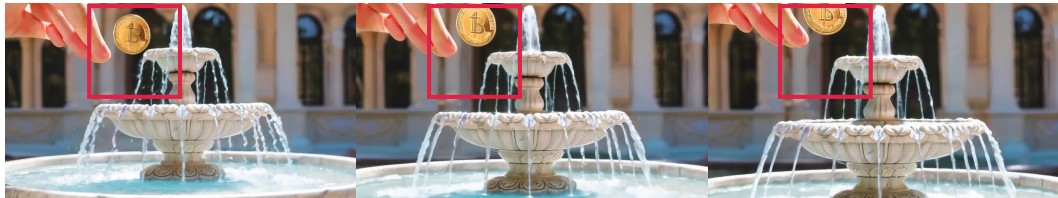

(b) *Coin flicking into a sparkling fountain*

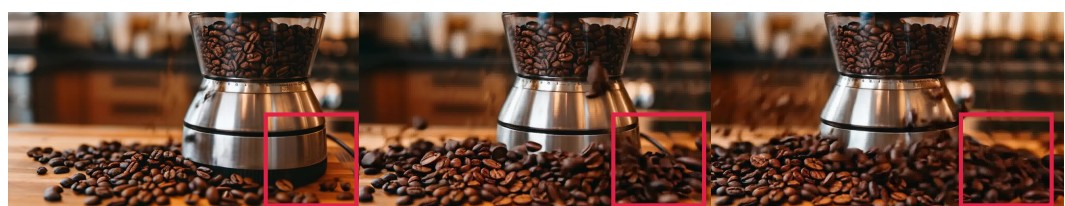

(c) *Metal grinder crushing coffee beans*

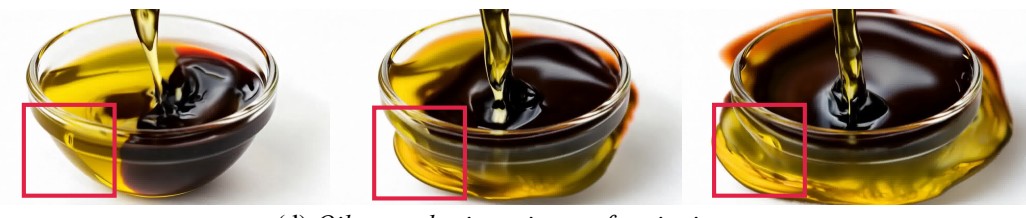

(d) *Oil cascades into vinegar for vinaigrette*

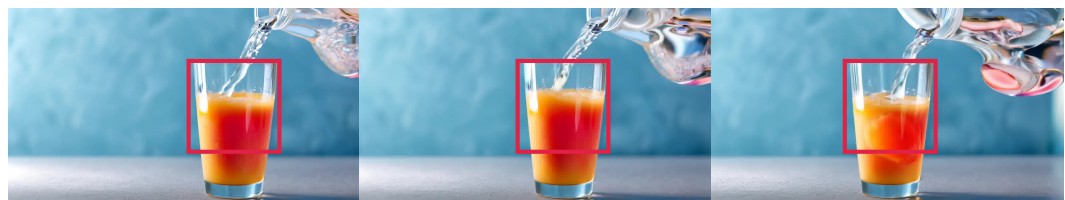

(e) *Water streams into fresh juice*

Figure 26: Unphysical Generated Examples of Dream Machine. (a) Newton's Second Law Violation: the diver floats in mid-air, defying gravity. (b) Newton's Second Law Violation: the coin hovers, disregarding gravitational forces. (c) Conservation of Mass Violation: numerous coffee beans appear spontaneously without a source. (d) Nonphysical Penetration: oil and vinegar pass through the glass cup. (e) Conservation of Mass Violation: the juice volume remains unchanged despite the addition of water.

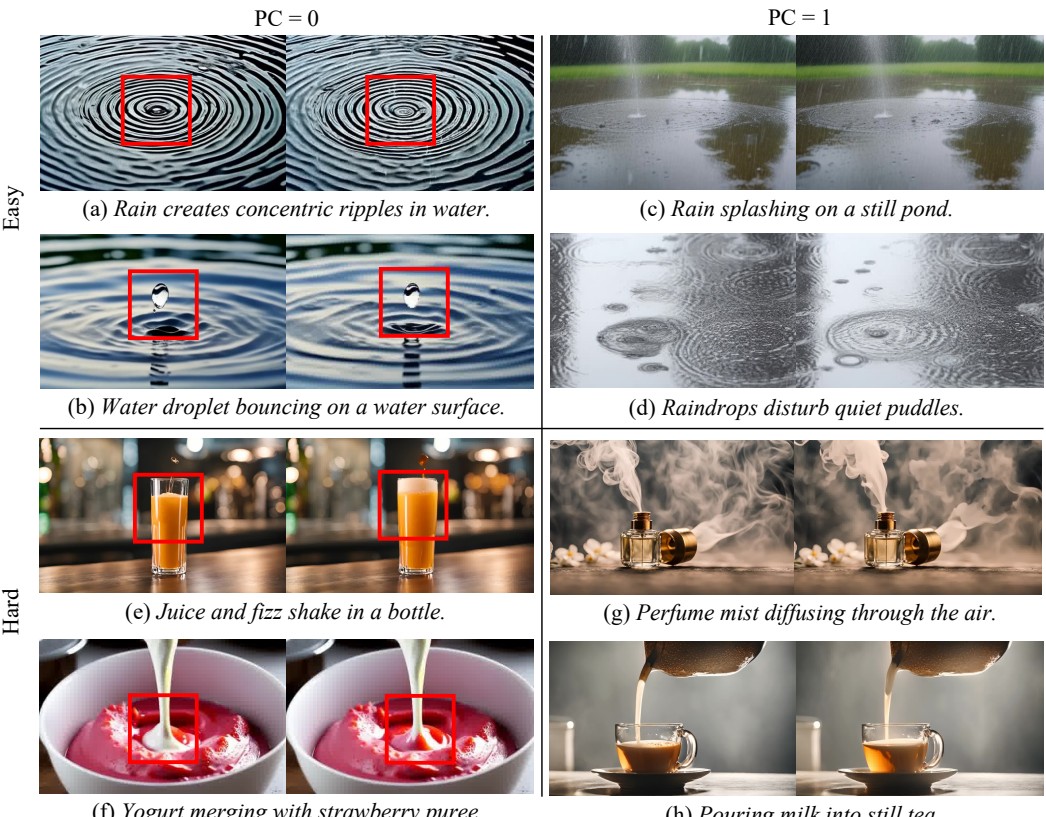

Figure 27: Qualitative examples in the fluid-fluid category. Videos in the left column have a majority PC score of 0, while videos in the right column have a majority PC score of 1. (a) The central ripple does not vanish even in absence of raindrops. (b) The water droplet is floating upwards, defying gravity. (e) The total volume of juice is increasing. (f) The color of the yogurt is not consistent over time.

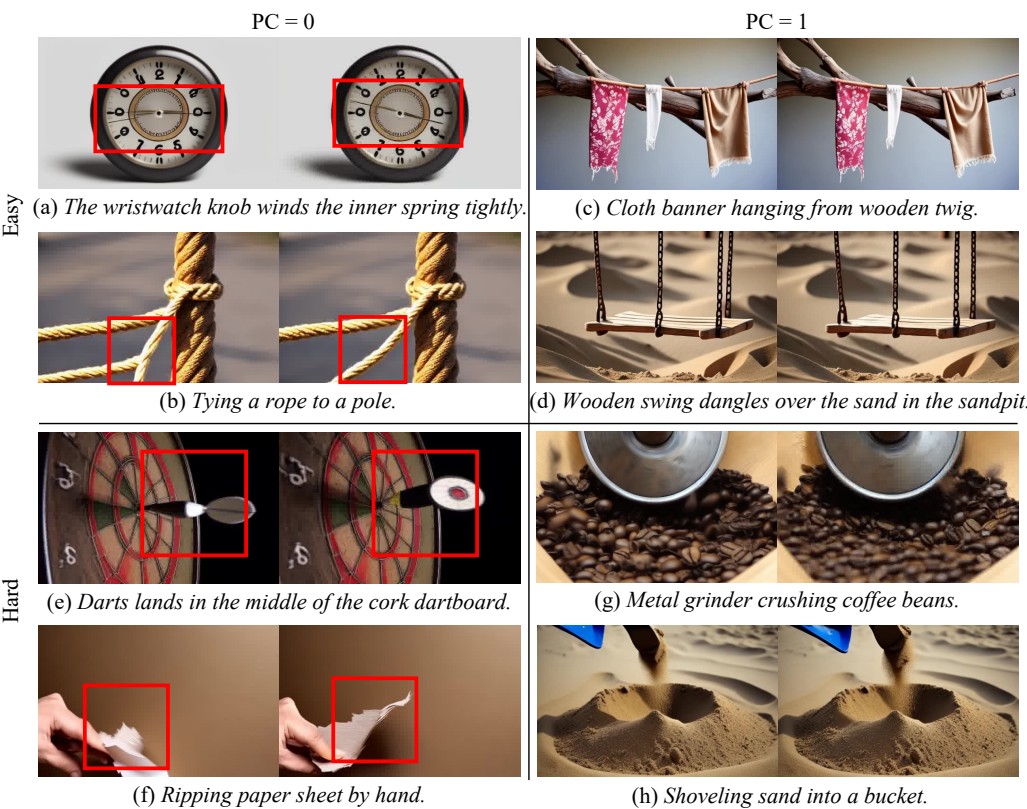

Figure 28: Qualitative examples in the solid-solid category. Videos in the left column have a majority PC score of 0, while videos in the right column have a majority PC score of 1. (a) The hands of the clock have illogical motion. (b) One piece of the robe disappears. (e) The geometry and texture of the dart are not consistent over time. (f) The total volume of the sheet of paper is not consistent over time.

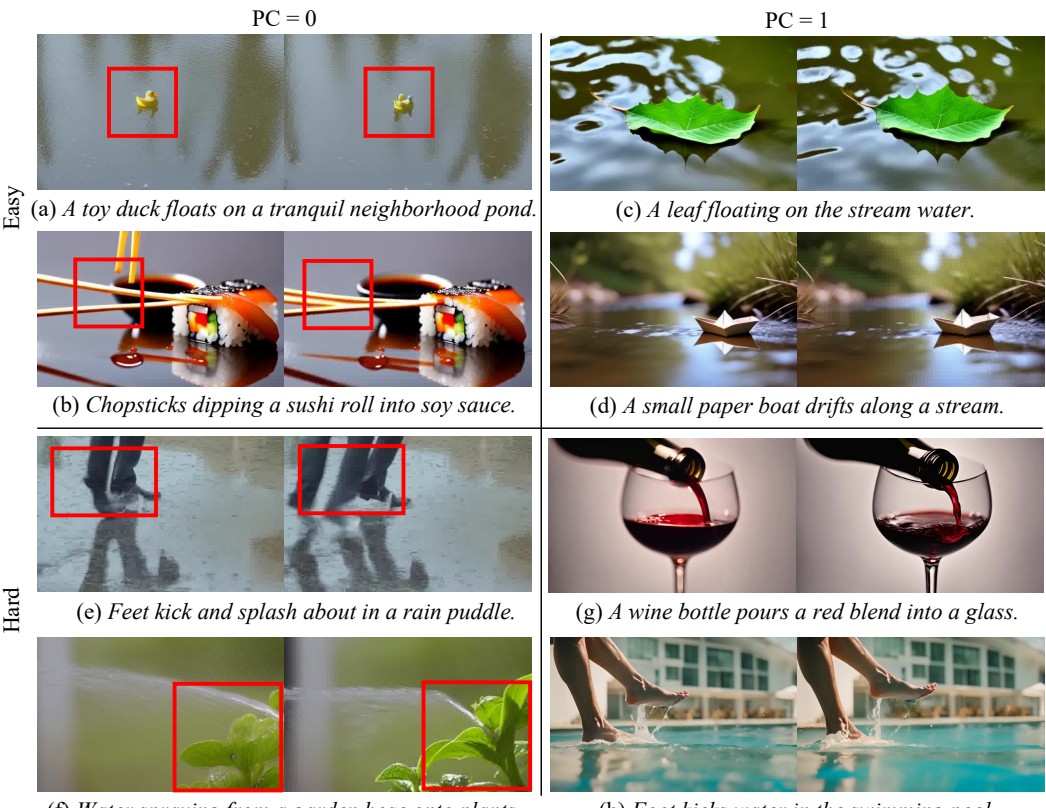

Figure 29: Qualitative examples in the fluid-fluid category. Videos in the left column have a majority PC score of 0, while videos in the right column have a majority PC score of 1. (a) The geometry and color of the duck head changes over time. (b) One chopstick appears from nowhere. (e) One leg appears from nowhere. (f) The geometry and texture of the leaves are not consistent over time.

