# OpenReview forum: "VideoPhy: Evaluating Physical Commonsense for Video Generation"
_ICLR.cc/2025/Conference — ICLR 2025 Poster_

### Official Review · Reviewer_dBNE · 2024-11-03

**Soundness:** 4
**Presentation:** 4
**Contribution:** 4
**Rating:** 8
**Confidence:** 4

**Summary:**

The paper presents a benchmark for evaluating physical commonsense for video generation. The benchmark includes 1) high-quality, human-verified captions; 2) automatic evaluation models to evaluate the video generation models; 3) the generated videos using existing methods and human annotations for these videos. Also, the authors reveal some conclusions based on the benchmark results, which could provide insights to the community.

**Strengths:**

1. The paper is well-written and easy to follow.
2. The paper presents a high-quality benchmark for evaluating physical commonsense for video generation. Careful data curation for text prompts is employed to ensure the quality of the text prompt.
3. The authors provide a comprehensive analysis of the text prompt used for evaluation. The category is balanced.
4. Beyond human evaluation, the authors also provide a model for automatic evaluation by fine-tuning a model for evaluating the physical commonsense of the model. The model eases the use of the benchmark for future research.
5. The authors also will release the generated videos and human evaluation for future research, which will potentially boost the performance of the automatic evaluation model.

**Weaknesses:**

1. For the VIDEOCON-PHYSICS model, have the authors conducted a human evaluation for the performance? Since the data used for training this model comes from same human annotators, the trained model may be biased. It would be better if the model can provide a human evaluation for the evaluation results generated by the VIDEOCON-PHYSICS model (just for the check of the correctness of the results provided by the VIDEOCON-PHYSICS model)

**Questions:**

Please see the weakness part for my concerns, but it is just a minor one.

---

> ### Author Response · Authors · 2024-11-16
> **Response to reviewer**
>
> We thank the reviewer for their encouraging comments. We are very excited to see that the reviewer finds our work (a) well-written and easy to follow, (b) high-quality benchmark for evaluating physical commonsense for video generation, (c) comprehensive in analysis, (d) reliable in terms of the automatic evaluation, and (e) relevant for boosting performance of automatic evaluation in the future.
>
> Q: Human evaluation of VideoCon-Physics
> - We clarify that Table 4 indicates the agreement (ROC-AUC) between the videocon-physics predictions and the human annotations on the unseen prompts and videos in the test set. The results highlight that the agreement of Videocon-physics is high in comparison to the other baselines.
> - Table 5 highlights that the VideoCon-physics is not biased towards annotations that were taken for specific video models. Specifically, we show that the Videocon-physics can reliably perform judgments for the video models that were unseen in the training on the unseen prompts.
> - Further, we highlight that the videophy annotations were performed by 14 workers where the load was shared uniformly across the annotators. It is unlikely that the model will be biased towards the judgments of a specific annotator.
> - Since the goal of automatic evaluation is to align with human judgment, we will publicly release the human annotations and automatic evaluation scores as a valuable resource for the community.

---

> > ### Comment · Reviewer_dBNE · 2024-11-26
> > **Comments after Rebuttal**
> >
> > Thanks for your clarification. The release of human annotations will definitely benefit the community! I will keep my original rating for accepting the paper.

---

### Official Review · Reviewer_xyoE · 2024-11-03

**Soundness:** 2
**Presentation:** 3
**Contribution:** 2
**Rating:** 5
**Confidence:** 2

**Summary:**

This paper introduces VideoPHY, a benchmark consisting of 688 captions designed to evaluate text-to-video models on physical commonsense. This work focuses on the real-world activities and interactions, classified by material interactions into three categories: solid-solid, solid-fluid, and fluid-fluid.

The dataset was initially generated using GPT-4 and then refined through manual filtering, and was annotated with binary (0/1) difficulty levels. Evaluation was based on two binary (0/1) metrics Semantic Adherence and Physical Commonsense, assessed by human or VLM. For the VLM, the authors fine-tuned VideoCon on human annotations, creating VideoCon-Physics.

Experiments comparing current SOTA text-to-video models including both open and closed models, indicate that current models struggle to model physical activities well.

**Strengths:**

1. This paper presents a physical commonsense benchmark that addresses a gap in existing datasets. The field of video generation needs such a benchmark, as t2v models gain popularity partly for their potential as world simulators or physical engines.
2. The dataset covers a wide range of activities, interactions, and dynamics on various materials as mentioned in line 194/195. The classification of dataset is somehow inspired by graphics field, and the difficulty of simulate those dynamics is also considered, which is a very interesting aspect for constructing the dataset.
3. Experiments are conducted over most of the current t2v models except for those without API support, providing a comprehensive comparison across a wide range of models.
4. This work provides insight and contributes to further improving t2v models.

**Weaknesses:**

1. I am not an expert in graphics or materials, so I am very uncertain about the category definitions and category ratios. I can get the idea of categorizing based on the state of matter, and solid and liquid are the most common. However, in graphics, rigid bodies, soft bodies, particle systems, fabrics, characters and animals are distinct topics that rely on very different physical models, whereas fluid dynamics, such as inviscid and viscous flows, are comparatively less diverse. If the idea of this benchmark is focusing on physical interactions and dynamics, then categorizing by interaction types or physical properties rather than broad material types can be more informative and nuanced. On the category ratios, as stated in line 179-181 as solids involve more physical constitutive models, we might expect more cases of solid-solid interactions than solid-fluid interactions, yet the sample counts are nearly equal (289 and 291, respectively).
2. Each sample in the training set for VideoCon-Physics was only labeled by one human annotator, while the author also mentioned while annotating benchmark, the agreement of three human annotators is 75% and 70% on SA and PC. This makes the training set less trustworthy.
3. I am also very skeptical about the motivation of fine-tuning VideoCon for direct evaluation of semantic adherence and physical commonsense. Those two metrics seem to cover a wide range of concepts but only one number between 0 and 1 was given. Directly fine-tuning VideoCon to evaluate especially the physical commonsense without introducing any extra knowledge, reasoning, or explicitly modeling does not make sense to me. The improvement of metrics might result from dataset domain bias, but there is no analysis on that. And from the leader boards of human evaluation and automatic evaluation, the disagreement on models such as Luma Dream Machine and Pika is not negligible. I suggest authors provide more detailed disagreement or evaluation statistics.

**Questions:**

see weaknesses above.

1. Given the challenges around categorization raised above, are there plans to expand or refine this benchmark to incorporate a broader range of interaction types or more nuanced physical properties, beyond basic material categories?
2. Have the authors considered using more detailed metrics to better decouple the concepts?

**Details Of Ethics Concerns:**

The authors mentioned the annotators might reflect perceptual biases. It would be nice to see a higher level of analysis on this.

---

> ### Author Response · Authors · 2024-11-16
> **Response to reviewer**
>
> We thank the reviewer for their diligent feedback. We are motivated to see that the reviewer finds our work (a) relevant as it addresses a gap in existing video evaluation datasets, (b) interesting in terms of dataset construction which is motivated from the computer graphics perspective, (c) comprehensive in conducting experiments for a wide range of T2V models, and (d) insightful for further improvements in T2V models.
>
> Q: Category definitions and category ratios
>
> - We believe it appropriate to group various types of solids into a single, unified 'solid' category, for our purpose. In theory, all solids can be prescribed as a universal constitutive model governed by the conservation of mass and momentum. Nevertheless, graphics researchers typically model certain materials using simplified constitutive models to lower computational costs. For instance, a rigid body, in reality, is a deformable body with a very high stiffness. In fact, exact rigidity does not exist in the real world. Similarly, researchers propose other solid constitutive models (e.g. fabrics, granular materials) to simplify computation for some specific material behaviors. Our choice of solid, as a broad categorization, serves as a basis for generalization beyond specific simulation techniques used in graphics research. Our benchmark is designed to capture a wide spectrum of physically plausible behaviors, rather than isolating specific graphics sub-domains.
> - Interactions between solid-solid pairs primarily focus on resolving contact constraints to prevent penetration. In contrast, solid-fluid interactions exhibit more diversity. For example, water can be repelled by an umbrella but can be absorbed by a sponge. Such behaviors, including permeability, adhesion, and absorption, are not available in solid-solid interactions. Given the diversity of solid-fluid interactions, we choose to balance the sample counts for solid-solid and solid-fluid interactions to ensure our benchmark does not overemphasize simpler contact-based interactions while underrepresenting more complex and varied solid-fluid dynamics.
> - We appreciate the reviewer's suggestion of categorizing based on interaction types or specific physical properties. This is an insightful point, and we agree that such an approach could provide additional granularity. However, our current choice prioritizes simplicity and broad applicability, especially for non-experts who may use the benchmark across various disciplines. Future work can consider more fine-grained categorization to better measure the ability of generative models to handle specific properties (e.g., plasticity, viscosity) if desired. We will add this discussion in the revised paper.
>
> Q: Training set of VideoCon-Physics
>
> - Since the training and testing set of the VideoPhy serve different purposes, we had to balance the resource allocation in the limited academic budget for human annotations. Specifically, the testing set was used to create a reliable leaderboard based on human judgements. In this case, we sample 1 video per test prompt and use three annotators to provide their judgements to ensure highest quality.
> - However, the role of the training set was to train a deep neural network based automatic evaluator which is more data hungry. Hence, we decided to sample 2 videos per train prompt which increases the diversity of the data, and got 1 annotator to judge it for text adherence and physical commonsense evaluation (12000 annotations in total).
> - We clarify that the task of semantic adherence and physical commonsense judgments is inherently subjective. Prior work such as ImageReward [1] or AlpacaEval [2] are widely adopted to study human preferences in generative models and achieve human agreement close to 65%. In this regard, our human agreements of 70%-75% are quite reasonable.
> - We respectfully disagree with the reviewer that the training set is not trustworthy. In fact, our empirical findings suggest that VideoCon-Physics achieves the highest agreement with the human judgments on the test set (Table 4). In addition, Table 5 shows that the VideoCon-Physics decisions align with the human judgements for unseen video models too. This would not have been possible if the training dataset was noisy.
> - Ideally, we agree that having more human judgements will benefit the data quality but it will significantly increase the data collection expenses, which goes beyond our budget. We will add this discussion in the limitations section (Appendix B).
>
> [1] ImageReward: https://arxiv.org/pdf/2304.05977 \
> [2] AlpacaEval: https://github.com/tatsu-lab/alpaca_eval

---

> ### Author Response · Authors · 2024-11-16
> **Response to reviewer (2/n)**
>
> Q: Binary feedback for evaluating semantic adherence and physical commonsense
>
> - We highlight that binary feedback (0/1) is quite popular in aligning generative models such as large language models [1]. Further, we observe that binary feedback is much easier to collect at industrial scale by big generative model providers (e.g., ChatGPT). For instance, we note that the ChatGPT user interface asks binary preference after generating the response to a simple query (https://ibb.co/6nHV85c). Similar extensions exist in the field of text-to-image generative models [2,3]. Hence, the binary feedback protocol is quite powerful in studying and improving the generative models.
> - The ability to assign a score to generated content is the common way to assess the model performance across various benchmarks [4,5]. In addition, it makes it easier for us to collect large-scale human annotations under a limited financial budget.
> - While the automatic evaluator is trained with the binary feedback, it can provide us with a continuous score between [0,1] which can be useful for fine-grained video assessment.
> - In addition, we agree with the reviewer that a dense feedback system would capture more nuanced mistakes of the video generative models (e.g., completing 8 movements versus 6 movements). However, designing such prompts is non-trivial, and evaluating the generated videos in such scenarios is much more challenging, labor-extensive, and expensive in the limited academic budget.
> - We firmly believe that binary feedback can provide a lot of interesting insights too. For instance, we uncovered the ability of the video generative models to perform differently for diverse material interactions (e.g., solid-solid, solid-fluid, fluid-fluid). In addition, we could gauge the model’s performance on easy and harder prompts too. Our qualitative evaluation confirms these differences observed in the quantitative values.
> - This work is intended to lay the foundation for physical commonsense so that the practitioners can compare existing models quantitatively. We believe that it will spark further research in various dimensions including the collection of diverse and denser forms of feedback. We will add this discussion explicitly in the revised paper.
>
> [1] Ethayarajh, K., Xu, W., Muennighoff, N., Jurafsky, D. and Kiela, D., 2024. Kto: Model alignment as prospect theoretic optimization. arXiv preprint arXiv:2402.01306. \
> [2] Li S, Kallidromitis K, Gokul A, Kato Y, Kozuka K. Aligning diffusion models by optimizing human utility. arXiv preprint arXiv:2404.04465. \
> [3] Lee, Kimin, et al. "Aligning text-to-image models using human feedback." arXiv preprint arXiv:2302.12192 (2023). \
> [4] Huang, Ziqi, Yinan He, Jiashuo Yu, Fan Zhang, Chenyang Si, Yuming Jiang, Yuanhan Zhang et al. "Vbench: Comprehensive benchmark suite for video generative models." In Proceedings of the IEEE/CVF Conference on Computer Vision and Pattern Recognition, pp. 21807-21818. 2024. \
> [5] Yarom, M., Bitton, Y., Changpinyo, S., Aharoni, R., Herzig, J., Lang, O., Ofek, E. and Szpektor, I., 2024. What you see is what you read? improving text-image alignment evaluation. Advances in Neural Information Processing Systems, 36.
>
>
> Q:  Fine-tuning VideoCon to evaluate especially the physical commonsense without introducing any extra knowledge, reasoning, or explicitly modeling.
>
> - In this work, we do not assume that the existing video-language models have physical commonsense understanding. Infact, our evaluation in Table 4 suggests that the existing models (Gemini-Pro-Vision and VideoCon) are very close to random (50) their agreements with human physical commonsense judgements.
> - To this end, we take a data-driven approach and finetune the model on 12K human annotations (L 350-351) to imbibe new knowledge about the semantic adherence and physical commonsense judgements with generated videos. Prior work [1,2] has shown that data-driven approaches can outperform rule-based physics simulators for more complicated systems like weather and climate.
> - We respectfully point out that finetuning the models with human annotations will come under the paradigm of introducing extra knowledge and explicit modeling.
> - In addition, we note that VideoCon-Physics should not be considered as a general-purpose physical commonsense evaluator. The purpose of this model is to allow fast evaluations of the videos generated by the prompts in the VideoPhy data. We believe that the road to building general-purpose physical commonsense evaluators is quite long as the field is in the nascent stages. We will add this discussion in the revised paper.
>
> [1] Climax: https://arxiv.org/abs/2301.10343 \
> [2] Stormer: https://arxiv.org/abs/2312.03876

---

> > ### Author Response · Authors · 2024-11-16
> > **Response to reviewer (3/n)**
> >
> > Q: Disagreement statistics between automatic and human leaderboard.
> >
> > - To address the reviewer’s comment, we perform more quantitative analysis to the data in Table 10. Specifically, we calculate the absolute rank difference between the human and automatic leaderboard for the open and closed models. We show the results below:
> >
> > | Human Ranking   |    Automatic Ranking           | Absolute rank difference |
> > |---------------|---------------|--------------------------|
> > |     Open models   |     |                          |
> > | CogVideoX-5B  | CogVideoX-5B  | 0                        |
> > | VideoCrafter2 | VideoCrafter2 | 0                        |
> > | CogVideoX-2B  | LaVIE         | 1                        |
> > | LaVIE         | CogVideoX-2B  | 1                        |
> > | SVD           | SVD           | 0                        |
> > | ZeroScope     | ZeroScope     | 0                        |
> > | OpenSora      | OpenSora      | 0                        |
> > | Closed models   |                |
> > | Pika          | Dream Machine | 3                        |
> > | Dream Machine | Lumiere-T2I2V | 1                        |
> > | Lumiere-T2I2V | Lumiere-T2V   | 1                        |
> > | Lumiere-T2V   | Pika          | 1                        |
> > | Gen-2         | Gen-2         | 0                        |
> > |               |               | Average = 0.66           |
> >
> > - Our analysis reveals that the average rank of the models in the automatic leaderboard is 0.66 above or below the expected rank of the model in the human leaderboard. This indicates VideoCon-Physics is reliable for evaluating the future models on our dataset. We agree with the difference between the rankings of Pika in the human and automatic leaderboard (also noted in L519-521). We believe that this can be fixed by acquiring more training data, which is an immediate future work.
> > - We will add the above quantitative analysis in the revised paper.
> >
> > Q: Refining the benchmark
> > - As mentioned by the reviewer in the strengths section, we note that the existing benchmark does cover a wide range of interactions and materials found in the real-world.
> > - We highlight that the vision of this project was to lay a solid foundation and start a dialogue amongst the practitioners to assess physical commonsense abilities. We are wholeheartedly devoted to expanding this dataset (as version 2.0) with several artifacts that build on the findings of this paper.
> >
> > Q: Detailed metrics to better decouple the concepts
> > - We respectfully point out that the existing benchmark provides several metrics and fine-grained analysis to understand the gaps in the modern video generative models.
> > - Firstly, we show that the models behave differently across diverse material interactions in Table 3. This is quite insightful to model builders as well as clients (e.g., film studios) who want to create specific kinds of motions.
> > - Secondly, our experiments in Table 6 suggest that video models struggle more on the harder captions than easier captions in both semantic and physical commonsense metrics. This highlights that the materials and motions that are harder to simulate with existing methods are also harder for video generative models to synthesize.
> > - Finally, we not only calculate marginal semantic adherence and physical commonsense but also look at their joint performance. We present the model performance across all possibilities {(SA,PC)=(1,1), (1,0), (0,1), (0,0)} in Appendix J.
> > - Despite these features, we are committed to adding more nuances (e.g., explanations) in the future versions of the benchmark, as discussed above.
> >
> >
> > Q: Perceptual bias of the annotators
> >
> > - As mentioned in the limitations section, our human annotators from the AMT platform belong to the US and Canada region. Prior works [1] have argued that the interpretation of visual content can differ across diverse cultures. For example, some cultures do not like “thumbs-up signals” while other cultures consider it as a simple gesture for approval. This perceptual bias is often used in consumer research for targeted marketing [2].
> > - In our context, perceptual biases can emerge in subtle ways. For example, there are various techniques for cooking fried rice, such as (a) stirring it with a large spoon or (b) tossing it in a wok [3]. For some annotators, the latter method may seem physically impractical depending on their cultural background.
> >
> > [1] Effect of culture on perception: https://core.ac.uk/download/pdf/16379016.pdf \
> > [2] https://www.youtube.com/watch?v=V-Pc-QUklQM&t=2s \
> > [3] https://www.youtube.com/watch?v=ywfBSnXklfk

---

> > > ### Author Response · Authors · 2024-11-20
> > > **Rebuttal Reminder**
> > >
> > > Hi,
> > >
> > > Thanks again for your insightful feedback on our work! We've carefully worked to address your comments/questions. Are there any further questions or concerns we should discuss?

---

> > > > ### Author Response · Authors · 2024-11-23
> > > > **Rebuttal Reminder 2**
> > > >
> > > > Hi,
> > > >
> > > > As the rebuttal round is coming to an end, it would be really helpful if we can address any of your additional comments/questions before that.

---

> > > > > ### Author Response · Authors · 2024-11-26
> > > > > **Rebuttal Reminder 3**
> > > > >
> > > > > Hi,
> > > > >
> > > > > We believe that we have addressed most of your concerns. Please let us know if we can address any of your additional comments/questions in the remaining time.

---

> > > > > > ### Comment · Reviewer_xyoE · 2024-11-26
> > > > > >
> > > > > > Thank you for your very detailed rebuttal also sorry for the late feedback. Generally, the authors well addressed my concern on category choices, training set, refining the benchmark, and perceptual bias.
> > > > > >
> > > > > > However, my concerns are:
> > > > > > - Binary feedback on only two fields are not strong enough for a benchmark. The author lists chatGPT and some literature [1,2,3] as binary feedback examples, which all use binary feedback as labels or additional information for fine-tuning. However, a benchmark should target towards more comprehensive and constructive comparisons.
> > > > > > - For automatic evaluation, I don't see a necessity on separating open and closed models rankings, and the differences suggest that the the automatic leaderboard is still not reliable enough. The author states in the rebuttal that Climax and Stormer use data-driven methods for complex systems like weather and climate, but a wide range of "physical commonsense" is a even more complicated concept. However, I acknowledge that there is not much better method at this point.
> > > > > >
> > > > > > From the above two points, if I stand in the position of proposing a new video generation method, I would consider this dataset a strong verification source, but I would still need to conduct a great amount of human evaluation to analyze how to improve my method as well as how to compare to others. Therefore overall, I think the paper has proposed a very good dataset and the insight is also very important to the community, but the proposed benchmark and evaluation is not very applicable at this point.
> > > > > >
> > > > > > I do agree with the contributions and the notable workload in this work, so I am raising my score to a borderline reject.

---

> > > > > > > ### Author Response · Authors · 2024-11-27
> > > > > > > **Response to reviewer**
> > > > > > >
> > > > > > > Hi,
> > > > > > >
> > > > > > > We thank the reviewer for the diligent feedback to our rebuttal.
> > > > > > >
> > > > > > > 1. We clarify that the binary feedback was collected across diverse categories of material interactions (Table 3) and hardness of the prompts (Table 6). In addition, we perform qualitative analysis of the generated videos and point out that the common failure modes include: (a) Conservation of mass violation: the volume or texture of an object is not consistent over time, (b) Newton’s First Law violation: an object changes its velocity in a balanced state without any external force, (c) Newton’s Second Law violation: an object violates the conversation of momentum, (d) Solid Constitutive Law violation: solids deform in ways that contradict their material properties, e.g., a rigid object deforming over time, (e) Fluid Constitutive Law violation: fluids exhibit unnatural flow motions, and (f) Non-physical penetration: objects unnaturally penetrate each other (L458-474).
> > > > > > > We hope this establishes the comprehensiveness and constructive insights in the dataset. We agree that more fine-grained evaluations will help in strengthening the scoring method, and we will expand these elements in the future versions of the data.
> > > > > > >
> > > > > > > 2.  We respectfully point out that average ranking difference of 0.66 is actually reasonable. Specifically, we bring the reviewer's attention to some of the popular works in the vision-language evaluation literature [1,2]. In Table 4 of [1], the mappings between human and automatic rankings do not match exactly. Similarly, Table 3 and Table 4 of [2] indicate that human and automatic rankings do not match for every model. We believe that VideoPhy serves as a strong foundation for future works on physical commonsense evaluation.
> > > > > > >
> > > > > > > [1] Vibe-Eval: https://arxiv.org/pdf/2405.02287 \
> > > > > > > [2] Visit-Bench: https://arxiv.org/pdf/2308.06595
> > > > > > >
> > > > > > > We thank the reviewer again for their feedback, and will be happy to answer any more questions that help in increasing your confidence in our work.

---

### Official Review · Reviewer_A9xM · 2024-11-04

**Soundness:** 3
**Presentation:** 3
**Contribution:** 4
**Rating:** 6
**Confidence:** 3

**Summary:**

This paper presents a benchmark designed to evaluate the physical commonsense of videos generated by text-to-video models. It highlights significant gaps in these models' ability to accurately simulate real-world physics and adhere to caption prompts. It introduces VideoPhy, a dataset consisting of real-world interaction prompts, and VideoCon-Physics, an automatic evaluation pipeline. Evaluation shows that even the best models, such as CogVideoX-5B, only achieve a 39.6% adherence rate to physical laws, emphasizing the need for improvements in video generation models.

**Strengths:**

* The paper shifts attention from general visual and semantic quality to the capability of T2V models to simulate real-world physics, addressing a vital aspect of realism in video generation.
* The paper provides detailed insights into different failure modes, guiding future model improvements and research directions.
* The automation pipeline, VideoCon-Physics enables scalable assessment of semantic adherence and physical commonsense in generated videos, which can be useful and meaningful to the research communities on T2V generation.

**Weaknesses:**

* Among the T2V models used for comparison, some still frequently fail to reproduce the scenarios specified by the text prompts. For example, in assessing physical reasoning in a scenario where milk is being poured, one needs to verify whether the milk appropriately fills the cup. However, in practice, these models often fail even to generate a video depicting the act of pouring milk. In such cases, the benchmark may be more influenced by the general video generation capabilities of the models rather than their physical commonsense reasoning abilities, as shown in the similar trends observed between Semantic Adherence and Physical Commonsense scores in Table 4. Therefore, it seems necessary to conduct experiments that assess physical commonsense only on generated videos that have appropriate semantic adherence, ensuring that the evaluation focuses on the models' understanding of physical phenomena rather than their basic ability to generate relevant videos.

* Proposed automating evaluations using a video-to-text model relies on a assumption: the V2T model must have a better understanding of physical phenomena than the T2V model. I remain some concerns about the justification of this assumption, as it is essential for the validity of the automated evaluation method. Without depending on this assumption, it is quite reasonable to suggest that the proposed V2T model can evaluate the T2V model because it has been fine-tuned on data containing these physical phenomena. In this context, I'm curious what it would be if the T2V model is similarly fine-tuned on a portion (training split) of the VideoPhy dataset and then generates videos based on prompts from the test split.

**Questions:**

Please see the weaknesses.

---

> ### Author Response · Authors · 2024-11-16
> **Response to the reviewer**
>
> We thank the reviewer for their insightful comments. We are happy to observe that the reviewer finds our work (a) vital in addressing the realism in video generation, (b) insightful in understanding the failure modes which are useful in guiding future model developments and research directions, and (c) useful and meaningful in terms of the automatic evaluation method.
>
> Q: Decoupling physical commonsense and semantic adherence.
>
> - We clarify that the physical commonsense score does not depend on the semantic adherence capability in our evaluation. As mentioned in Section 3.2 and 3.3, the human and automatic evaluator do not focus on the underlying caption to make the physical commonsense judgements.
> - Ideally, we want the models to follow the prompt and generate physically commonsensical videos. To this end, we study the joint performance (SA=1, PC=1) in our main results (Figure 1, L256-258).
> - In Table 3, the first column provides the joint performance (SA=1,PC=1), marginal semantic adherence (SA=1) and marginal physical commonsense (PC=1). A reader can estimate the posterior performance (PC=1 given SA=1) by taking the ratio of the joint performance and marginal semantic adherence scores. By default, we do not report posterior performance since it can be inferred from the existing numbers. In addition, just the posterior performance does not provide the entire picture which is clearer with joint performance metric.
> - Finally, we believe that a bad model can easily game the posterior metric. For example, a bad model can generate a video which aligns with the prompt for 1 out of 700 prompts in the dataset. Now, assume that this video is also accurate in terms of physical commonsense. Hence, the posterior performance of this model will be 100%. This can be quite misleading for the practitioners.
> - We present the model performance across all possibilities {(SA,PC)=(1,1), (1,0), (0,1), (0,0)} in Appendix J. We will add this discussion explicitly in the revised paper.
>
> Q: Assumption in the automatic evaluation
> - In this work, we do not assume that the existing V2T model has physical commonsense understanding. Infact, our evaluation in Table 4 suggests that the existing models (Gemini-Pro-Vision and VideoCon) are very close to random (50) their agreements with human physical commonsense judgements.
> - To this end, we take a data-driven approach and finetune the model on 12K human annotations (L 350-351) to imbibe new knowledge about the semantic adherence and physical commonsense judgements with generated videos.
> - Since the goal of automatic evaluation is to align with human judgment, we will publicly release the human annotations and automatic evaluation scores as a valuable resource for the community.
>
> Q: Finetuning video generative models on training split of VideoPhy
>
> - In Appendix Q, we finetune Lumiere-T2I2V with the training split of VideoPhy. Specifically, we train it with the videos in our training dataset which achieve a score of 1 on physical commonsense and a score of 1 on semantic adherence. In total, there are ~1000 such videos. While this dataset is small for finetuning, we perform a finetuning run to address the reviewer’s comment. Post-finetuning, we generate the videos for the test prompts and evaluate them using our automatic evaluator:
> | Model                    | SA   | PC   | Average |
> |--------------------------|------|------|---------|
> | Lumiere-T2I2V-Pretrained | 46   | 25   | 35      |
> | Lumiere-T2I2V-Finetuned  | 36.5 | 24.6 | 30.5    |
> - We find that the semantic adherence (video-text alignment) reduces by a large margin and physical commonsense remains unchanged after finetuning. This can be due to several factors: (a) the number of training samples is not enough, (b) optimization difficulties since the training videos are generated from several generative models (mix of on-policy and off-policy data), and (c) vanilla finetuning being a bad algorithm for learning from these samples. Since post-training of video generative models is a less explored direction, there can be many ways to improve the generative model’s physical commonsense.
> - Further, we clarify that improving video generative models is an entirely new project. We ran the experiment in Appendix Q to inspire future research on enhancing physical commonsense in generated videos (L525-530).
> - These results also show that mere finetuning with the samples in the training set of VideoPhy does not lead to large gains in the automatic evaluation on the test set. We will add this discussion to the revised paper.

---

> > ### Comment · Reviewer_A9xM · 2024-11-19
> >
> > Thank the authors for answering the questions and providing additional results. I also thank the authors for pointing out some parts I missed in the paper. After reading the authors' responses and revisiting the paper, some of my major concerns have been addressed. I am raising my rating.

---

> > > ### Author Response · Authors · 2024-11-20
> > > **Thanks**
> > >
> > > Thank you for increasing your rating. If we have clarified your major concerns, please consider adjusting your soundness scores too. Also, feel free to ask if you have more questions.

---

### Official Review · Reviewer_tYi4 · 2024-11-04

**Soundness:** 3
**Presentation:** 3
**Contribution:** 3
**Rating:** 6
**Confidence:** 4

**Summary:**

This work targets at building a benchmark that can evaluate the physical commonsense for video generation. Multiple physics-related prompts are first generated and evaluated, serving as the input for different video generators. With human evaluation, it is found that both open-source and closed-source, and find that they significantly lack physical commonsense and semantic adherence capabilities. In order to evaluate in scale, an auto-evaluator is trained.

**Strengths:**

The goal of evaluating physical commonsense is quite important for this area.
The chosen video generators are comprehensive, including open-source models and closed ones.
The presentation is well-organized and easy to follow, with key numbers regarding evaluations.

**Weaknesses:**

Regarding evaluation, I have several major concerns.

- As mentioned in Sec.3.1, binary feedback (0/1) is used to evaluate semantic adherence and physical commonsense. This discrete value may not reflect and moniter the true capability for different video generators. For example, for a text prompt with 10 physical movements, one generator achieves 8 movements while another is 6. These binary feedback can not tell the gap between two candidates. This example could be too extreme while that could be a weakness of binary value.
- Besides, I am not sure whether the absolute accuracy of physical achievements is a proper metric.  Especially for Fig.1, I believe the relative scores across different generators (like ELO score) make more sense, which also avoid the weakness of binary feedback
- Regarding physical commonsense (PC), it really depends on the text following ability of given generators (semantic adherence in this work).  Joint performance may be one alternative for both text and physics evaluation while the posterior probability may be one perspective for physical commonsense alone.

For auto-evaluator videocon-physics, the fine-tuning details could facilitate the reproduction and transferability from VIDEOCON to other video-language models.

**Questions:**

Please see Weaknesses.

---

> ### Author Response · Authors · 2024-11-20
> **Response to reviewer**
>
> We thank the reviewer for their diligent feedback. We are motivated to find that the reviewer finds our work: (a) quite important for this area, (b) comprehensive in coverage of open and close video generative models, (c) well-organized and easy to follow. We address the reviewer comments below:
>
> Q: Binary Feedback
>
> - We highlight that binary feedback (0/1) is quite popular in aligning generative models such as large language models [1]. Further, we observe that binary feedback is much easier to collect at industrial scale by big generative model providers (e.g., ChatGPT). For instance, we note that the ChatGPT user interface asks binary preference after generating the response to a simple query (https://ibb.co/6nHV85c). Similar extensions exist in the field of text-to-image generative models [2,3]. Hence, the binary feedback protocol is quite powerful in studying and improving the generative models.
> - In addition, we agree with the reviewer that a dense feedback system would capture more nuanced mistakes of the video generative models (e.g., completing 8 movements versus 6 movements). However, designing such prompts is non-trivial, and evaluating the generated videos in such scenarios is much more challenging, labor-extensive, and expensive (esp. with limited academic budgets).
> - We also believe that binary feedback can provide a lot of interesting insights too. For instance, we uncovered the ability of the video generative models to perform differently for diverse material interactions (e.g., solid-solid, solid-fluid, fluid-fluid). In addition, we could gauge the model’s performance on easy and harder prompts too. Our qualitative evaluation confirms these differences observed in the quantitative values.
> - While the automatic evaluator is trained with the binary feedback, it can provide us with a continuous score between [0,1] which can be useful for fine-grained video assessment.
> - This work is intended to lay the foundation for physical commonsense so that the practitioners can compare existing models quantitatively. We believe that it will spark further research in various dimensions including the collection of diverse and denser forms of feedback. We will add this discussion explicitly in the revised paper.
>
> [1] Ethayarajh, K., Xu, W., Muennighoff, N., Jurafsky, D. and Kiela, D., 2024. Kto: Model alignment as prospect theoretic optimization. arXiv preprint arXiv:2402.01306. \
> [2] Li S, Kallidromitis K, Gokul A, Kato Y, Kozuka K. Aligning diffusion models by optimizing human utility. arXiv preprint arXiv:2404.04465. \
> [3] Lee, Kimin, et al. "Aligning text-to-image models using human feedback." arXiv preprint arXiv:2302.12192 (2023).
>
> Q: Rankings vs. binary feedback
> - To address the reviewer’s comment, we have performed a new ranking-based study for physical commonsense evaluation. Specifically, we ask the three workers to look at two videos simultaneously and pick the one with better physical commonsense. In particular, we got 500 pairwise comparisons for 4 video generative models (CogVideoX-5B, Pika, Gen2, OpenSora). It costs us $360 to run this human eval. Subsequently, we computed the ELO scores of these models based on the human annotations. We present the results below:
>
> | Model        | PC ELOScore[New] | PC Binary %[Existing paper] |
> |--------------|------------------|-----------------------------|
> | CogVideoX-5B | 1081             | 53                          |
> | Pika         | 1048             | 36.5                        |
> | Gen2         | 1010             | 27.2                        |
> | OpenSora     | 860              | 23.5                        |
>
> - Interestingly, we find that the relative ranking of these models remains unchanged under both the feedback methods. Specifically, CogVideoX-5B and OpenSora are still the best and worst models on the VideoPhy dataset, respectively. We will add these results to the revised paper.
> - We note that the open (usually smaller) video generative models will be penalized for losing to close (usually larger) video generative models in the ranking-based setup. The absolute feedback operates independently across all video generative models, and helps in better contextualizing the capability of the models with similar scales. Anecdotally, practitioners do not look at Lmsys (chat arena) ELO leaderboard to understand the capabilities of small language models because they are usually at the bottom of that list after losing to strong models in many comparisons. For example, GPT-4/Gemini models have saturated the MATH dataset but it is still used as a guiding star for others who want to build strong models with math capabilities. We diligently believe that such revolution can be sparked by VideoPhy for the field of video generative models.

---

> > ### Author Response · Authors · 2024-11-20
> > **Response to reviewer (2/n)**
> >
> > Q: Decoupling physical commonsense and semantic adherence.
> >
> > - We clarify that the physical commonsense score does not depend on the semantic adherence capability in our evaluation. As mentioned in Section 3.2 and 3.3, the human and automatic evaluator do not focus on the underlying caption to make the physical commonsense judgements.
> > - Ideally, we want the models to follow the prompt and generate physically commonsensical videos. To this end, we study the joint performance (SA=1, PC=1) in our main results (Figure 1, L256-258).
> > - In Table 3, the first column provides the joint performance (SA=1,PC=1), marginal semantic adherence (SA=1) and marginal physical commonsense (PC=1). A reader can estimate the posterior performance (PC=1 given SA=1) by taking the ratio of the joint performance and marginal semantic adherence scores. By default, we do not report posterior performance since it can be inferred from the existing numbers. In addition, just the posterior performance does not provide the entire picture which is clearer with joint performance metric.
> > - Finally, we believe that a bad model can easily game the posterior metric. For example, a bad model can generate a video which aligns with the prompt for 1 out of 700 prompts in the dataset. Now, assume that this video is also accurate in terms of physical commonsense. Hence, the posterior performance of this model will be 100%. This can be quite misleading for the practitioners.
> > - We present the model performance across all possibilities {(SA,PC)=(1,1), (1,0), (0,1), (0,0)} in Appendix J. We will add this discussion explicitly in the revised paper.
> >
> > Q: VideoconPhy reproducibility
> >
> > - As mentioned in the reproducibility statement, we provide the finetuning details for VideoPhy in Appendix M.
> > - In addition, we plan to release the model checkpoint, data and finetuning code for transferability to other video-language models.

---

> > > ### Author Response · Authors · 2024-11-23
> > > **Reminder for the reviewer**
> > >
> > > Hi,
> > >
> > > Thanks again for your insightful feedback on our work! We've carefully worked to address your comments/questions. Are there any further questions or concerns we should discuss?

---

> > > > ### Author Response · Authors · 2024-11-26
> > > > **Rebuttal Reminder 2**
> > > >
> > > > Hi,
> > > >
> > > > We believe that we have addressed most of your concerns. Please let us know if we can address any of your additional comments/questions in the remaining time.

---

> > > ### Comment · Reviewer_tYi4 · 2024-11-26
> > >
> > > Thanks for your response. My major concerns are addressed. Thus I lean to accept.
> > >
> > > I strongly recommend authors could further discuss the model performance across all possibilities in revision.

---

> > > > ### Author Response · Authors · 2024-11-27
> > > > **Response to Reviewer**
> > > >
> > > > Hi,
> > > >
> > > > We thank the reviewer for their feedback and increasing their score. Feel free to ask more questions if it helps in increasing your confidence in our work.

---

### Author Response · Authors · 2024-11-25
**Revised paper update**

We have uploaded the revised version of the paper which addresses most of the comments from the reviewers (highlighted in blue).

---

### Meta-Review · Area_Chair_XNVG · 2024-12-19

**Metareview:**

The paper addresses an important gap in evaluating physical commonsense in text-to-video (T2V) generation models. The proposed VideoPhy benchmark is insightful, covering diverse physical interactions and revealing significant shortcomings in current models. Reviewers appreciated the comprehensive evaluation, clear presentation, and the automation pipeline VideoCon-Physics. Concerns include the limitations of binary feedback, potential biases in annotations, and the need for more nuanced metrics. The authors' revisions addressed these issues effectively, leading to improved reviewer ratings. Overall, this work is a valuable contribution to the community. The AC recommends acceptance.

**Additional Comments On Reviewer Discussion:**

Reviewers raised concerns about binary feedback, decoupling semantic adherence and physical commonsense, and biases in the automatic evaluator. The authors justified binary feedback's practicality, clarified metrics, provided new ranking experiments, and addressed annotation bias by releasing data. Revisions included detailed explanations and new analyses, satisfying most concerns. While some limitations remain, the work's insights and benchmark value outweigh these, led to the final decision to accept.

---

### Decision · Program_Chairs · 2025-01-22

Accept (Poster)